# Unbalanced Low-rank Optimal Transport Solvers

**Meyer Scetbon**[*]
Microsoft Research
t-mscetbon@microsoft.com

**Michael Klein**[*]
Apple
michalk@apple.com

**Giovanni Palla**
Helmholtz Center Munich
giovanni.palla@helmholtz-muenchen.de

**Marco Cuturi**
Apple
cuturi@apple.com

## Abstract

Two salient limitations have long hindered the relevance of optimal transport methods to machine learning. First, the $O(n^3)$ computational cost of standard sample-based solvers (when used on batches of $n$ samples) is prohibitive. Second, the mass conservation constraint makes OT solvers too rigid in practice: because they must match *all* points from both measures, their output can be heavily influenced by outliers. A flurry of recent works has addressed these computational and modeling limitations, but has resulted in two separate strains of methods: While the computational outlook was much improved by entropic regularization, more recent $O(n)$ linear-time *low-rank* solvers hold the promise to scale up OT further. In terms of modeling flexibility, the rigidity of mass conservation has been eased for entropic regularized OT, thanks to unbalanced variants of OT that can penalize couplings whose marginals deviate from those specified by the source and target distributions. The goal of this paper is to merge these two strains, low-rank and unbalanced, to achieve the promise of solvers that are *both* scalable and versatile. We propose custom algorithms to implement these extensions for the linear OT problem and its fused-Gromov-Wasserstein generalization, and demonstrate their practical relevance to challenging spatial transcriptomics matching problems. These algorithms are implemented in the `ott-jax` toolbox [Cuturi et al., 2022].

## 1 Introduction

Recent machine learning (ML) works have witnessed a flurry of activity around optimal transport (OT) methods. The OT toolbox provides convenient, intuitive and versatile ways to quantify the difference between two probability measures, either to quantify a distance (the Wasserstein and Gromov-Wasserstein distances), or, in more elaborate scenarios, by computing a push-forward map that can transform one measure into the other [Peyré and Cuturi, 2019]. Recent examples include, e.g., single-cell omics [Bunne et al., 2021, 2022, Demetci et al., 2020, Nitzan et al., 2019, Cang et al., 2023, Klein et al., 2023], attention mechanisms [Tay et al., 2020, Sander et al., 2022], self-supervised learning[Caron et al., 2020, Oquab et al., 2023], and learning on graphs [Vincent-Cuaz et al., 2023].

**On the challenges of using OT.** Despite their long history in ML [Rubner et al., 2000], OT methods have long suffered from various limitations, that arise from their statistical, computational, and modelling aspects. The *statistical* argument is commonly referred to as the curse-of-dimensionality of OT estimators: the Wasserstein distance between two probability densities, and its associated optimal Monge map, is poorly approximated using samples as the dimension $d$ of observation grows [Dudley et al., 1966, Boissard and Le Gouic, 2014]. On the *computational* side, computing OT between a pair of $n$ samples involves solving a (generalized) matching problem, with a price of $O(n^3)$ and above [Kuhn, 1955, Ahuja et al., 1993]. Finally, the original *model* for OT rests on a

37th Conference on Neural Information Processing Systems (NeurIPS 2023).

mass conservation constraint: all observations from either samples must be accounted for, including outliers that are prevalent in machine learning datasets. Combined, these weaknesses have long hindered the use of OT, until a more recent generation of solvers addressed these three crucial issues.

**The Entropic Success Story.** The winning approach, so far, to carry out that agenda has been entropic regularization methods [Cuturi, 2013]. The computational virtues of the Sinkhorn algorithm when solving OT [Altschuler et al., 2017, Peyré et al., 2016, Solomon et al., 2016, Le et al., 2021] come with statistical efficiency [Genevay et al., 2019, Mena and Niles-Weed, 2019, Chizat et al., 2020], and can also be seamlessly combined with *unbalanced* formulations by penalizing – rather than constraint – mass conservation, both for the linear [Frogner et al., 2015, Chizat et al., 2018, Séjourné et al., 2022, Fatras et al., 2021, Pham et al., 2020] and quadratic [Séjourné et al., 2021] problems. These developments have all been implemented in popular OT packages [Feydy et al., 2019, Flamary et al., 2021, Cuturi et al., 2022].

**The Low-Rank Alternative.** A recent strain of solvers relies instead on *low-rank* (LR) properties of cost and coupling matrices [Forrow et al., 2018, Scetbon and Cuturi, 2020, Scetbon et al., 2021]. Much like entropic solvers, these LR solvers have a better statistical outlook [Scetbon and Cuturi, 2022] and extend to GW problems [Scetbon et al., 2022]. In stark contrast to entropic solvers, however, LR solvers benefit from linear complexity $O(nrd)$ w.r.t sample size $n$ (using rank $r$ and cost dimension $d$) that can scale to ambitious tasks where entropic solvers fail [Klein et al., 2023].

**The Need for Unbalanced Low-Rank Solvers.** LR solvers do suffer, however, from a major practical limitation: their inability to handle unbalanced problems. Yet, unbalancedness is a crucial ingredient for OT to be practically relevant. This is exemplified by the fact that unbalancedness played a crucial role in the seminal reference [Schiebinger et al., 2019], where it is used to model cell birth and death.

**Our Contributions** We propose in this work to lift this last limitation for LR solvers to:

- Incorporate unbalanced regularizers to define a LR linear solver (§ 3.1);
- Provide accelerated algorithms, inspired by some of the recent corrections proposed by [Séjourné et al., 2022], to isolate translation terms that appear in dual subroutines (§ 3.2);
- Carry over and adapt these approaches to the GW (§ 3.3) and Fused-GW problems (§ 3.4);
- Carry out an exhaustive hyperparameter selection procedure within large scale OT tasks (spatial transcriptomics, brain imaging), and demonstrate the benefits of our approach (§ 4).

## 2   Reminders on Low-Rank Transport and Unbalanced Transport

We consider two metric spaces $(\mathcal{X}, d_{\mathcal{X}})$ and $(\mathcal{Y}, d_{\mathcal{Y}})$, as well as a cost function $c : \mathcal{X} \times \mathcal{Y} \to [0, +\infty[$. The simplex $\Delta_n^+$ holds all positive $n$-vectors summing to 1. For $n, m \geq 1, a \in \Delta_n^+$, and $b \in \Delta_m^+$, given points $x_1, \ldots, x_n \in \mathcal{X}$ and $y_1, \ldots, y_m \in \mathcal{Y}$, we define two discrete probability measures $\mu$ and $\nu$ as $\mu := \sum_{i=1}^n a_i \delta_{x_i}, \nu := \sum_{j=1}^m b_j \delta_{y_j}$ where $\delta_z$ is the Dirac mass at $z$.

**Cost matrices.** For $q \geq 1$, consider first two square pairwise *cost* matrices, each encoding the geometries of points *within* $\mu$ and $\nu$, and a rectangular matrix that studies that *across* their support:

$$A := [d_{\mathcal{X}}^q(x_i, x_{i'})]_{1 \leq i, i' \leq n}, \ B := [d_{\mathcal{Y}}^q(y_j, y_{j'})]_{1 \leq j, j' \leq m}, \ C := [c(x_i, y_j)]_{\substack{1 \leq i \leq n, \\ 1 \leq j \leq m}}.$$

**The Kantorovich Formulation of OT** is defined as the following parameterized linear program:

$$\mathrm{OT}(\mu, \nu) := \min_{P \in \Pi_{a,b}} \langle C, P \rangle, \quad \text{where} \quad \Pi_{a,b} := \left\{ P \in \mathbb{R}_+^{n \times m}, \text{ s.t. } P\mathbf{1}_m = a, \ P^T \mathbf{1}_n = b \right\}. \quad (1)$$

**The Low-Rank Formulation of OT** is best understood as a variant of (1) that rests on a low-rank *property* for cost matrix $C$, and low-rank *constraints* for couplings $P$. More precisely, Scetbon et al. [2021] propose to constraint the set of admissible couplings to those, within $\Pi_{a,b}$, that have a non-negative rank of $r \geq 1$. That set can be equivalently reparameterized as

$$\Pi_{a,b}(r) = \{P \in \mathbb{R}_+^{n \times m} | P = Q \operatorname{diag}(1/g) R^T, \ Q \in \Pi_{a,g}, \ R \in \Pi_{b,g}, \text{ and } g \in \Delta_r^+\}.$$

The low-rank optimal transport (LOT) problem simply uses that restriction in (1) to define :

$$\mathrm{LOT}_r(\mu, \nu) := \min_{P \in \Pi_{a,b}(r)} \langle C, P \rangle = \min_{Q \in \Pi_{a,g}, R \in \Pi_{a,g}, g \in \Delta_r^+} \langle C, Q \operatorname{diag}(1/g) R \rangle. \quad (2)$$

Scetbon et al. [2021] propose and prove the convergence of a mirror-descent scheme to solve (2), and obtain linear time and memory complexities with respect to the number of samples, where each iteration in that descent scales as $(n + m)rd$, where $d$ is the rank of $C$.

**The Unbalanced Formulation of OT** starts from (1) as well, but proposes to do without $\Pi_{a,b}$ and its marginal constraints [Frogner et al., 2015, Chizat et al., 2018], and rely instead on two regularizers:

$$\text{UOT}(\mu, \nu) := \min_{P \in \mathbb{R}_+^{n \times m}} \langle C, P \rangle + \tau_1 \text{KL}(P\mathbf{1}_m | a) + \tau_2 \text{KL}(P^T \mathbf{1}_n | b) \tag{3}$$

where $\tau_1, \tau_2 > 0$ and $\text{KL}(p|q) := \sum_i p_i \log(p_i/q_i) + q_i - p_i$. This formulation is solved using entropic regularization, with modified Sinkhorn updates [Frogner et al., 2015]. *Proposing an efficient algorithm able to merge* (2) *with* (3) *is the first goal of this paper.*

**Gromov-Wasserstein (GW) Considerations.** The GW problem [Mémoli, 2011] is a generalization of (1) where the energy $\mathcal{Q}_{A,B}$ is a quadratic function of $P$ defined through inner cost matrices $A$, $B$:

$$\mathcal{Q}_{A,B}(P) := \sum_{i,j,i',j'} (A_{ii'} - B_{jj'})^2 P_{ij} P_{i'j'} = \mathbf{1}_m^T P^T A^{\odot 2} P \mathbf{1}_m + \mathbf{1}_n^T P B^{\odot 2} P^T \mathbf{1}_n - 2\langle APB, P \rangle \tag{4}$$

where $\odot$ is the Hadamard product. To minimize (4), the default approach rests on entropic regularization [Solomon et al., 2016, Peyré et al., 2016] and variants [Sato et al., 2020, Blumberg et al., 2020, Xu et al., 2019, Li et al., 2023]. Scetbon et al. [2022] adapted the low-rank framework to minimize $\mathcal{Q}_{A,B}$ over low-rank matrices $P$, achieving a linear-time complexity when $A$ and $B$ are themselves low-rank. Independently, [Séjourné et al., 2021] proposed an unbalanced generalization that also applies to GW and which can be implemented practically using entropic regularization. Finally, the minimization of a composite objective involving the sum of $\mathcal{Q}_{A,B}$ with $\langle C, \cdot \rangle$ is known as the *fused* GW problem [Vayer et al., 2018].

## 3 Unbalanced Low-Rank Transport

### 3.1 Unbalanced Low-rank Linear Optimal Transport

We incorporate unbalancedness to low-rank solvers [Scetbon et al., 2021, 2022], moving gradually from the linear problem to the more involved GW and FGW problem. Using the framework of [Frogner et al., 2015, Chizat et al., 2018], we extend first the definition of LOT, introduced in (2), to the unbalanced case by considering the following optimization problem:

$$\text{ULOT}_r(\mu, \nu) := \min_{P:\, \text{rk}_+(P) \leq r} \langle C, P \rangle + \tau_1 \text{KL}(P\mathbf{1}_m | a) + \tau_2 \text{KL}(P^T \mathbf{1}_n | b), \tag{5}$$

where $\text{rk}_+(P)$ denotes the non-negative rank of $P$. Therefore by denoting $\Pi_r := \{(Q, R, g) \in \mathbb{R}_+^{n \times r} \times \mathbb{R}_+^{m \times r} \times \mathbb{R}_+^r : Q^T \mathbf{1}_n = R^T \mathbf{1}_m = g\}$, and using the reparameterization of low-rank couplings, we obtain the following equivalent formulation of ULOT:

$$\text{ULOT}_r(\mu, \nu) = \min_{(Q,R,g) \in \Pi_r} \underbrace{\langle C, Q \operatorname{diag}(1/g) R^T \rangle}_{\mathcal{L}_C(Q,R,g)} + \underbrace{\tau_1 \text{KL}(Q\mathbf{1}_r | a) + \tau_2 \text{KL}(R\mathbf{1}_r | b)}_{\mathcal{G}_{a,b}(Q,R,g)} . \tag{6}$$

We introduce the more compact notation $\mathcal{G}_{a,b}(Q, R, g) := F_{\tau_1,a}(Q\mathbf{1}_r) + F_{\tau_2,b}(R\mathbf{1}_r)$, where $F_{\tau,z}(s) := \tau \text{KL}(s|z)$ for $\tau > 0$ and $z \geq 0$ coordinate-wise. To solve (6), and using this split, we move away from mirror-descent and apply instead proximal gradient-descent for the KL divergence. At each iteration, we consider a linear approximation of $\mathcal{L}_C$ where a KL penalization is added to the objective (as in the classical mirror descent scheme). However, we leave $\mathcal{G}_{a,b}$ intact at each iteration. Borrowing notations from [Scetbon et al., 2021], we must solve at each iteration the convex optimization problem:

$$(Q_{k+1}, R_{k+1}, g_{k+1}) := \underset{\zeta \in \Pi_r}{\text{argmin}} \frac{1}{\gamma_k} \text{KL}(\zeta, \xi_k) + \tau_1 \text{KL}(Q\mathbf{1}_r | a) + \tau_2 \text{KL}(R\mathbf{1}_r | b), \tag{7}$$

where $(Q_0, R_0, g_0) \in \Pi_r$ is the initialization, and the triplet $\xi_k := (\xi_k^{(1)}, \xi_k^{(2)}, \xi_k^{(3)})$ holds synthetic costs matrices that are re-computed at each iteration $k$:

$$\xi_k^{(1)} := Q_k \odot e^{-\gamma_k C R_k \operatorname{diag}(1/g_k)}, \xi_k^{(2)} := R_k \odot e^{-\gamma_k C^T Q_k \operatorname{diag}(1/g_k)}, \xi_k^{(3)} := g_k \odot e^{\gamma_k \omega_k / g_k^2},$$

with $[\omega_k]_i := [Q_k^T C R_k]_{i,i} \in \mathbb{R}^r$, and $(\gamma_k)_{k\geq 0}$ is a sequence of positive step sizes.

**Reformulation using Duality.** To solve (7), we apply Dykstra's algorithm [1983], whose iterations correspond to an alternating maximization on the dual formulation of (7):

**Proposition 1.** *The convex optimization problem defined in (7) admits the following dual:*

$$\sup_{f_1, h_1, f_2, h_2} \mathcal{D}_k(f_1, h_1, f_2, h_2) := -F^\star_{\tau_1, a}(-f_1) - \frac{1}{\gamma_k} \langle e^{\gamma_k(f_1 \oplus h_1)} - 1, \xi_k^{(1)} \rangle$$
$$- F^\star_{\tau_2, b}(-f_2) - \frac{1}{\gamma_k} \langle e^{\gamma_k(f_2 \oplus h_2)} - 1, \xi_k^{(2)} \rangle - \frac{1}{\gamma_k} \langle e^{-\gamma_k(h_1 + h_2)} - 1, \xi_k^{(3)} \rangle \tag{8}$$

*where $h_1, h_2 \in \mathbb{R}^r$, $f_1 \in \mathbb{R}^n$, $f_2 \in \mathbb{R}^m$, $F^\star_{\tau,z}(\cdot) := \sup_y \{\langle y, \cdot \rangle - F_{\tau,z}(y)\}$ is the convex conjugate of $F_{\tau,z}$. In addition strong duality holds and the primal problem admits a unique minimizer.*

**Remark 1.** *While we stick to KL regularizers in this work for simplicity, it is worth noting that this can be extended to more generic regularizers $F_{\tau_1,a}$ and $F_{\tau_2,b}$, as considered by Chizat et al. [2018].*

We use an alternating maximization scheme to solve (8). Starting from $h_1^{(0)} = h_2^{(0)} = \mathbf{0}_r$, we apply for $\ell \geq 0$ the following updates (dropping iteration number $k$ in (7) for simplicity):

$$f_1^{(\ell+1)} := \arg\sup_z \mathcal{D}(z, h_1^{(\ell)}, f_2^{(\ell)}, h_2^{(\ell)}), \ f_2^{(\ell+1)} := \arg\sup_z \mathcal{D}(f_1^{(\ell+1)}, h_1^{(\ell)}, z, h_2^{(\ell)}),$$
$$(h_1^{(\ell+1)}, h_2^{(\ell+1)}) := \arg\sup_{z_1, z_2} \mathcal{D}(f_1^{(\ell+1)}, z_1, f_2^{(\ell+1)}, z_2).$$

These maximizations can all be obtained in closed form, to result in the closed-form updates:

$$\exp(\gamma f_1^{(\ell+1)}) = \left( \frac{a}{\xi^{(1)} \exp(\gamma h_1^{(\ell)})} \right)^{\frac{\tau_1}{\tau_1 + 1/\gamma}}, \quad \exp(\gamma f_2^{(\ell+1)}) = \left( \frac{b}{\xi^{(2)} \exp(\gamma h_2^{(\ell)})} \right)^{\frac{\tau_2}{\tau_2 + 1/\gamma}}$$
$$g_{\ell+1} := \left( \xi^{(3)} \odot (\xi^{(1)})^T \exp(\gamma f_1^{(\ell+1)}) \odot (\xi^{(2)})^T \exp(\gamma f_2^{(\ell+1)}) \right)^{1/3}$$
$$\exp(\gamma h_1^{(\ell+1)}) = \frac{g_{\ell+1}}{(\xi^{(1)})^T \exp(\gamma f_1^{(\ell+1)})}, \quad \exp(\gamma h_2^{(\ell+1)}) = \frac{g_{\ell+1}}{(\xi^{(2)})^T \exp(\gamma f_2^{(\ell+1)})}$$

When using "scaling" representations for these dual variables, $\ell \geq 0$, $u_i^{(\ell)} := \exp(\gamma f_i^{(\ell)})$ and $v_i^{(\ell)} := \exp(\gamma h_i^{(\ell)})$ for $i \in \{1, 2\}$, we obtain a simple update, provided in the appendix (Alg. 5).

**Initialization and Termination.** We use the stopping criterion proposed in [Scetbon et al., 2021] to terminate the algorithm, $\Delta(\zeta, \tilde{\zeta}, \gamma) := \frac{1}{\gamma^2}(\mathrm{KL}(\zeta, \tilde{\zeta}) + \mathrm{KL}(\tilde{\zeta}, \zeta))$. Finding an efficient initialization for that problem is challenging, and various choices have been implemented for instance in [Cuturi et al., 2022]. We adopt the practical choices proposed in [Scetbon and Cuturi, 2022], using either random subcoupling matrices or a $k$-means approach, and also follow them in adapting the choice of $\gamma_k$ at each iteration $k$ of the outer loop. We summarize our proposal in Algorithm 1, which can be seen as an extension of [Scetbon et al., 2021, Alg.2].

**Convergence.** The convergence proof for Dykstra's algorithm, as implemented in Alg. 5 (see appendix), follows from [Bauschke and Combettes, 2008]). Scetbon et al. [2021] show the convergence of their scheme towards a stationary point, w.r.t to the criterion $\Delta(\cdot, \cdot, \gamma)$, for fixed $\gamma$. The stationary convergence of our proposed algorithm can be directly derived from their result.

**Complexity.** Given $\boldsymbol{\xi}$, solving Eq. (7) requires a time and memory complexity of $\mathcal{O}((n + m)r)$. However computing $\boldsymbol{\xi}$ requires in general $\mathcal{O}((n^2 + m^2)r)$ time and $\mathcal{O}(n^2 + m^2)$ memory. Scetbon et al. [2021] propose to consider low-rank factorizations of the cost matrix $C$ of the form $C \simeq C_1 C_2^T$ where $C_1 \in \mathbb{R}^{n \times d}$ and $C_2 \in \mathbb{R}^{m \times d}$. In that case computing $\boldsymbol{\xi}$ can be done in $\mathcal{O}((n + m)rd)$ time and $\mathcal{O}((n + m)(r + d))$ memory. Such factorizations are either known explicitly (e.g. when using squared-Euclidean distances) or can be obtained as approximations using the algorithm in [Indyk et al., 2019], which guarantees that for any distance matrix $C \in \mathbb{R}^{n \times m}$ and $\alpha > 0$ it can output matrices $C_1 \in \mathbb{R}^{n \times d}$, $C_2 \in \mathbb{R}^{m \times d}$ in $\mathcal{O}((m + n)\mathrm{poly}(\frac{d}{\alpha}))$ algebraic operations such that with probability at least 0.99, $\|C - C_1 C_2^T\|_F^2 \leq \|C - C_d\|_F^2 + \alpha\|C\|_F^2$, where $C_d$ denotes the best rank-$d$ approximation to $C$.

**Algorithm 1** ULOT($C, a, b, r, \gamma_0, \tau_1, \tau_2, \delta$)

---

**Inputs:** $C, a, b, r, \gamma_0, \tau_1, \tau_2, \delta$
$Q, R, g \leftarrow$ Initialization as proposed in [Scetbon and Cuturi, 2022]
**repeat**
$\quad$ $\tilde{Q} = Q, \ \tilde{R} = R, \ \tilde{g} = g,$
$\quad$ $\nabla_Q = CR \operatorname{diag}(1/g), \ \nabla_R = C^\top Q \operatorname{diag}(1/g),$
$\quad$ $\omega \leftarrow \mathcal{D}(Q^T CR), \ \nabla_g = -\omega/g^2,$
$\quad$ $\gamma \leftarrow \gamma_0 / \max(\|\nabla_Q\|_\infty^2, \|\nabla_R\|_\infty^2, \|\nabla_g\|_\infty^2),$
$\quad$ $\xi^{(1)} \leftarrow Q \odot \exp(-\gamma \nabla_Q), \ \xi^{(2)} \leftarrow R \odot \exp(-\gamma \nabla_R), \ \xi^{(3)} \leftarrow g \odot \exp(-\gamma \nabla_g),$
$\quad$ $Q, R, g \leftarrow$ ULR-Dykstra($a, b, \boldsymbol{\xi}, \gamma, \tau_1, \tau_2, \delta$) (Alg. 5)
**until** $\Delta((Q, R, g), (\tilde{Q}, \tilde{R}, \tilde{g}), \gamma) < \delta$;
**Result:** $Q, R, g$

---

### 3.2 Improvements on the Unbalanced Dykstra Algorithm

A well documented source of instability of unbalanced formulations of OT lies in the fact that the total mass of the optimal unbalanced coupling is not known beforehand. Séjourné et al. [2022] have proposed a technique to address this issue, with the benefit of reduced computational costs. They propose first a dual objective that is *translation* invariant (TI). We take inspiration from this strategy and adapt it to our problem, to propose the following variant of (8):

$$\sup_{\tilde{f}_1, \tilde{h}_1, \tilde{f}_2, \tilde{h}_2} \left( \mathcal{D}_{\text{TI}}(\tilde{f}_1, \tilde{h}_1, \tilde{f}_2, \tilde{h}_2) := \sup_{\lambda_1, \lambda_2 \in \mathbb{R}} \mathcal{D}(\tilde{f}_1 + \lambda_1, \tilde{h}_1 - \lambda_1, \tilde{f}_2 + \lambda_2, \tilde{h}_2 - \lambda_2) \right) \quad (9)$$

It is clear from the reparameterization that both problems (8) and (9) have the same value and also that $(\tilde{f}_1, \tilde{h}_1, \tilde{f}_2, \tilde{h}_2)$ is solution of (9) if and only if $(\tilde{f}_1 + \lambda_1^\star, \tilde{h}_1 - \lambda_1^\star, \tilde{f}_2 + \lambda_2^\star, \tilde{h}_2 - \lambda_2^\star)$ is solution of (8) where $(\lambda_1^\star, \lambda_2^\star)$ solves $\mathcal{D}_{\text{TI}}(\tilde{f}_1, \tilde{h}_1, \tilde{f}_2, \tilde{h}_2)$. To solve (9), we show that the variational formulation of the translation invariant dual objective targeted inside (9) can be obtained in closed form.

**Proposition 2.** *Let $\tilde{f}_1 \in \mathbb{R}^n$, $\tilde{f}_2 \in \mathbb{R}^m$ and $\tilde{h}_1, \tilde{h}_2 \in \mathbb{R}^r$, then the inner problem defined in (9) by $\mathcal{D}_{TI}(\tilde{f}_1, \tilde{h}_1, \tilde{f}_2, \tilde{h}_2)$ admits a unique solution $(\lambda_1^\star, \lambda_2^\star)$ and we have that*

$$\lambda_1^\star := \left( 1 - \frac{\tau_1 \tau_2}{(1/\gamma + \tau_1)(1/\gamma + \tau_2)} \right)^{-1} \left( \frac{\tau_1/\gamma}{1/\gamma + \tau_1} c_1 - \frac{\tau_1/\gamma}{1/\gamma + \tau_1} \frac{\tau_2}{1/\gamma + \tau_2} c_2 \right) \quad (10)$$

$$\lambda_2^\star := \left( 1 - \frac{\tau_1 \tau_2}{(1/\gamma + \tau_1)(1/\gamma + \tau_2)} \right)^{-1} \left( \frac{\tau_2/\gamma}{1/\gamma + \tau_2} c_2 - \frac{\tau_1/\gamma}{1/\gamma + \tau_1} \frac{\tau_2}{1/\gamma + \tau_2} c_1 \right) \quad (11)$$

*where*

$$c_1 := \log \left( \frac{\langle \exp(-\tilde{f}_1/\tau_1), a \rangle}{\langle \exp(-\gamma(\tilde{h}_1 + \tilde{h}_2)), \xi^{(3)} \rangle} \right), \quad \text{and} \quad c_2 := \log \left( \frac{\langle \exp(-\tilde{f}_2/\tau_2), a \rangle}{\langle \exp(-\gamma(\tilde{h}_1 + \tilde{h}_2)), \xi^{(3)} \rangle} \right).$$

Using Proposition 2, we perform an alternate maximization scheme on the TI formulation of the dual $\mathcal{D}_{\text{TI}}$. Indeed using Danskin's theorem (under the assumption that $\lambda_1^\star, \lambda_2^\star$ do not diverge), one obtains a variant of Algorithm 5. That TI approach is summarized below in Algorithm 3, using Algorithm 2 as a subroutine. We show in the experiments section (**Exp. 1**) that the TI approach has better computational performance on a simple task.

---

**Algorithm 2** compute-lambdas($a, b, \xi^{(3)}, u_1, v_1, u_2, v_2, \gamma, \tau_1, \tau_2$)

---

**Inputs:** $a, b, \xi^{(3)}, u_1, v_1, u_2, v_2, \gamma, \tau_1, \tau_2$
$\tilde{u}_1 \leftarrow u_1^{-1/\gamma/\tau_1}, \ \tilde{u}_2 \leftarrow u_2^{-1/\gamma/\tau_2}$
$c_1 \leftarrow \log(\langle \tilde{u}_1, a \rangle) - \log(\langle \xi^{(3)}, v_1^{-1} \odot v_2^{-1} \rangle), \ c_2 \leftarrow \log(\langle \tilde{u}_2, b \rangle) - \log(\langle \xi^{(3)}, v_1^{-1} \odot v_2^{-1} \rangle)$
**Result:** $\lambda_1^\star, \ \lambda_2^\star$ as in (10), (11)

---

---

**Algorithm 3** ULR-TI-Dykstra$(a, b, \boldsymbol{\xi}, \gamma, \tau_1, \tau_2, \delta)$

---

**Inputs:** $a, b, \boldsymbol{\xi} = (\xi^{(1)}, \xi^{(2)}, \xi^{(3)}), \gamma, \tau_1, \tau_2, \delta$
$v_1 = v_2 = \mathbf{1}_r, u_1 = \mathbf{1}_n, u_2 = \mathbf{1}_m$
**repeat**

$\quad$ $\tilde{v}_1 = v_1, \tilde{v}_2 = v_2, \tilde{u}_1 = u_1, \tilde{u}_2 = u_2$

$\quad$ $\lambda_1, \lambda_2 \leftarrow$ compute-lambdas$(a, b, \xi^{(3)}, u_1, v_1, u_2, v_2, \gamma, \tau_1, \tau_2)$ (Alg. 2)

$\quad$ $u_1 = \left(\frac{a}{\xi^{(1)}v_1}\right)^{\frac{\tau_1}{\tau_1 + 1/\gamma}} \exp(-\lambda_1/\tau_1)^{\frac{1}{1/\gamma + \tau_1}}, \quad u_2 = \left(\frac{b}{\xi^{(2)}v_2}\right)^{\frac{\tau_2}{\tau_2 + 1/\gamma}} \exp(-\lambda_2/\tau_2)^{\frac{1}{1/\gamma + \tau_2}},$

$\quad$ $\lambda_1, \lambda_2 \leftarrow$ compute-lambdas$(a, b, \xi^{(3)}, u_1, v_1, u_2, v_2, \gamma, \tau_1, \tau_2)$ (Alg. 2)

$\quad$ $g = \exp(\gamma(\lambda_1 + \lambda_2))^{1/3} \left(\xi^{(3)} \odot (\xi^{(1)})^T u_1 \odot (\xi^{(2)})^T u_2\right)^{1/3}, v_1 = \frac{g}{(\xi^{(1)})^T u_1}, v_2 = \frac{g}{(\xi^{(2)})^T u_2}$

**until** $\frac{1}{\gamma} \max(\|\log(u_i/\tilde{u}_i)\|_\infty, \|\log(v_i/\tilde{v}_i)\|_\infty) < \delta$;

**Result:** $\operatorname{diag}(u_1)\xi_k^{(1)}\operatorname{diag}(v_1), \quad \operatorname{diag}(u_2)\xi_k^{(2)}\operatorname{diag}(v_2), \quad g$

---

### 3.3 Unbalanced Low-rank Gromov-Wasserstein

The low-rank Gromov-Wasssertein (LGW) problem [Scetbon et al., 2022] between the two discrete metric measure spaces $(\mu, d_{\mathcal{X}})$ and $(\nu, d_{\mathcal{Y}})$, written for compactness using $(a, A)$ and $(b, B)$, reads

$$\mathrm{LGW}_r((a, A), (b, B)) = \min_{P \in \Pi_{a,b}(r)} \mathcal{Q}_{A,B}(P), \tag{12}$$

Building upon § 3.1 and leveraging the TI variant presented in § 3.2, we introduce the unbalanced low-rank Gromov-Wasserstein (ULGW) problem. There is, however, a significant challenge that appears when introducing unbalanced regularizers in (12): When $P$ is constrained to be in $\Pi_{a,b}$, the first two terms of the RHS in (12) simplify to $a^T A^{\odot 2} a + b^T B^{\odot 2} b$. Hence, they are constant and discarded when optimizing. In an unbalanced setting, these terms vary and must be accounted for:

$$\mathrm{ULGW}_r((a, A), (b, B)) := \min_{(Q,R,g) \in \Pi_r} \langle A^{\odot 2} Q\mathbf{1}_r, Q\mathbf{1}_r \rangle + \langle B^{\odot 2} R\mathbf{1}_r, R\mathbf{1}_r \rangle$$
$$- 2\langle AQ\operatorname{diag}(1/g)R^T B, Q\operatorname{diag}(1/g)R^T \rangle + \tau_1 \mathrm{KL}(Q\mathbf{1}_r|a) + \tau_2 \mathrm{KL}(R\mathbf{1}_r|b) \tag{13}$$

To solve the problem, we apply the same scheme as proposed for ULOT, that is a proximal gradient descent where we linearize $\mathcal{Q}_{A,B}$ and add a KL penalization while leaving the soft marginal constraints unchanged. Therefore the algorithm to solve ULGW is the same as that solving ULOT, however, the kernels $\boldsymbol{\xi}_k$ now take into account the quadratic terms of the original LGW problem. More formally, at each iteration $k$ of the outer loop, we propose to solve

$$(Q_{k+1}, R_{k+1}, g_{k+1}) := \underset{\boldsymbol{\zeta} \in \Pi_r}{\operatorname{argmin}} \frac{1}{\gamma_k} \mathrm{KL}(\boldsymbol{\zeta}|\boldsymbol{\xi}_k) + \tau_1 \mathrm{KL}(Q\mathbf{1}_r|a) + \tau_2 \mathrm{KL}(R\mathbf{1}_r|b), \tag{14}$$

where $(Q_0, R_0, g_0) \in \Pi_r$ is the initialization, $(\gamma_k)_{k \geq 0}$ a sequence of positive step sizes. Using notation $P_k = Q_k \operatorname{diag}(1/g_k)R_k^T$, the synthetic cost matrices $\boldsymbol{\xi}_k := (\xi_k^{(1)}, \xi_k^{(2)}, \xi_k^{(3)})$ are updated as:

$$\xi_k^{(1)} := Q_k \odot \exp(-2\gamma_k A^{\odot 2} Q_k \mathbf{1}_r \mathbf{1}_r^T) \odot \exp(-4\gamma_k A P_k B R_k \operatorname{diag}(1/g_k))),$$
$$\xi_k^{(2)} := R_k \odot \exp(-2\gamma_k B^{\odot 2} R_k \mathbf{1}_r \mathbf{1}_r^T) \odot \exp(-4\gamma_k B P_k^T A Q_k \operatorname{diag}(1/g_k))),$$
$$\xi_k^{(3)} := g_k \odot \exp(4\gamma_k \omega_k/g_k^2) \quad \text{with} \quad [\omega_k]_i := [Q_k^T A P_k B R_k]_{i,i} \in \mathbb{R}^r.$$

Note that (14) is the exact same optimization problem as (7), where only $\boldsymbol{\xi}_k$ has changed and therefore can be solved using Algorithm 3. Algorithm 4 summarizes our strategy to solve (13).

---

**Algorithm 4** $\text{ULGW}(A, B, a, b, r, \gamma_0, \tau_1, \tau_2, \delta)$

---

**Inputs:** $A, B, a, b, r, \gamma_0, \tau_1, \tau_2, \delta$
$Q, R, g \leftarrow$ Initialization as proposed in [Scetbon and Cuturi, 2022]
**repeat**

$\quad \tilde{Q} = Q, \ \tilde{R} = R, \ \tilde{g} = g,$
$\quad \nabla_Q = 4AQ \operatorname{diag}(1/g) R^T BR \operatorname{diag}(1/g) + 2A^{\odot 2} Q \mathbf{1}_r \mathbf{1}_r^T,$
$\quad \nabla_R = 4BR \operatorname{diag}(1/g) Q^T AQ \operatorname{diag}(1/g) + 2B^{\odot 2} R \mathbf{1}_r \mathbf{1}_r^T,$
$\quad \omega \leftarrow \mathcal{D}(Q^T AQ \operatorname{diag}(1/g) R^T BR), \ \nabla_g = -\omega/g^2,$
$\quad \gamma \leftarrow \gamma_0/\max(\|\nabla_Q\|_\infty^2, \|\nabla_R\|_\infty^2, \|\nabla_g\|_\infty^2),$
$\quad \xi^{(1)} \leftarrow Q \odot \exp(-\gamma \nabla_Q), \ \xi^{(2)} \leftarrow R \odot \exp(-\gamma \nabla_R), \ \xi^{(3)} \leftarrow g \odot \exp(-\gamma_k \nabla_g),$
$\quad Q, R, g \leftarrow \text{ULR-TI-Dykstra}(a, b, \boldsymbol{\xi}, \gamma, \tau_1, \tau_2, \delta)$ (Alg. 3)

**until** $\Delta((Q, R, g), (\tilde{Q}, \tilde{R}, \tilde{g}), \gamma) < \delta$;
**Result:** $Q, R, g$

---

**Convergence and Complexity.** Similarly to linear ULOT, the unbalanced Dykstra algorithm is guaranteed to converge [Bauschke and Lewis, 2000]. Because we use Algorithm 5, we retain exactly the same complexity, both in terms of time of memory, to solve these inner problems. The slight variation in kernel $\boldsymbol{\xi}$ compared to ULOT still retains the same $\mathcal{O}((n^2 + m^2)r)$ time and $\mathcal{O}(n^2 + m^2)$ memory complexities. However, as in ULOT, we can take advantage of low-rank approximations of *both* costs matrices $A$ and $B$ to reach linear complexity. Indeed, assuming $A \simeq A_1 A_2^T$ and $B \simeq B_1 B_2$ where $A_1, A_2 \in \mathbb{R}^{n \times d_X}$ and $B_1, B_2 \in \mathbb{R}^{m \times d_Y}$, then the total time and memory complexities become respectively $\mathcal{O}(mr(r + d_Y) + nr(r + d_X))$ and $\mathcal{O}((n + m)(r + d_X + d_Y))$. Again, when $A$ and $B$ are distance matrices, we use the algorithms from [Indyk et al., 2019].

## 3.4 Unbalanced Low-rank Fused-Gromov-Wasserstein

We finally focus on the increasingly impactful [Klein et al., 2023] fused-Gromov-Wasserstein problem, which merges linear and quadratic objectives [Vayer et al., 2018]:

$$\text{FGW}(\mu, \nu) := \min_{P \in \Pi_{a,b}} \alpha \langle C, P \rangle + \bar{\alpha} \mathcal{Q}_{A,B}(P) \tag{15}$$

where $\alpha \in [0, 1]$ and $\bar{\alpha} := 1 - \alpha$ allows interpolating between the GW and linear OT geometries. This problem remains a GW problem, where one replaces the 4-way cost $M[i, i', j, j'] := (A_{i,i'} - B_{j,j'})^2$ appearing in (4) by a composite interpolated cost between the OT and GW geometries, redefined as $M[i, i', j, j'] = \alpha C_{i,j} + \bar{\alpha}(A_{i,i'} - B_{j,j'})^2$. Our proposed unbalanced and low-rank version of the FGW problem includes $|P| := \|P\|_1$ the mass of $P$, to homogenize linear and quadratic terms,

$$\text{ULFGW}_r(\mu, \nu) := \min_{P: \, \text{rk}_+(P) \leq r} \alpha |P| \langle C, P \rangle + \bar{\alpha} \mathcal{Q}_{A,B}(P) + \tau_1 \text{KL}(P \mathbf{1}_m | a) + \tau_2 \text{KL}(P^T \mathbf{1}_n | b), \tag{16}$$

which is expanded through the explicit factorization of $P$, noticing that $|P| = |g| := \|g\|_1$:

$$\text{ULFGW}_r(\mu, \nu) := \min_{(Q, R, g) \in \Pi_r} \alpha |g| \mathcal{L}_C(Q, R, g) + \bar{\alpha} \mathcal{Q}_{A,B}(Q, R, g) + \mathcal{G}_{a,b}(Q, R, g) \tag{17}$$

Then by linearizing again $\mathcal{H} : (Q, R, g) \to \alpha |g| \mathcal{L}_C(Q, R, g) + \bar{\alpha} \mathcal{Q}_{A,B}(Q, R, g)$ with an added KL penalty and leaving $\mathcal{G}_{a,b}$ unchanged, we obtain at each iteration, the same optimization problem as in (14) where the kernels $\boldsymbol{\xi}_k$ are now defined as

$$\boldsymbol{\xi}_k := (\xi_k^{(1)}, \xi_k^{(2)}, \xi_k^{(3)}),$$

$$\xi_k^{(1)} := Q_k \odot \exp(-\gamma_k \nabla_Q \mathcal{H}_k), \ \xi_k^{(2)} := R_k \odot \exp(-\gamma_k \nabla_Q \mathcal{H}_k), \ \xi_k^{(3)} := g_k \odot \exp(-\gamma_k \nabla_g \mathcal{H}_k)$$

$$\nabla_Q \mathcal{H}_k := \alpha |g_k| CR_k \operatorname{diag}(1/g_k) + \bar{\alpha} \left(2A^{\odot 2} Q_k \mathbf{1}_r \mathbf{1}_r^T + 4AP_k BR_k \operatorname{diag}(1/g_k)\right)$$

$$\nabla_R \mathcal{H}_k := \alpha |g_k| C^T Q_k \operatorname{diag}(1/g_k) + \bar{\alpha} \left(2B^{\odot 2} R_k \mathbf{1}_r \mathbf{1}_r^T + 4BP_k^T AQ_k \operatorname{diag}(1/g_k)\right)$$

$$\nabla_g \mathcal{H}_k := \alpha \left(\langle C, P_k \rangle \mathbf{1}_r - |g_k| \omega_k^{\text{lin}}/g_k^2\right) - 4\bar{\alpha} \omega_k^{\text{quad}}/g_k^2$$

$$[\omega_k^{\text{lin}}]_i := [Q_k^T CR_k]_{i,i}, \ [\omega_k^{\text{quad}}]_i := [Q_k^T AP_k BR_k]_{i,i} \ \forall i \in \{1, \ldots, r\}.$$

These steps are summarized in Alg. 6 (see appendix). These steps result in a quadratic complexity, both in time and memory, with respect to the number of points $n$ and $m$. However, these complexities become *linear* when square matrices $A, B$ *and* rectangular $C$ all admit a low-rank factorization.

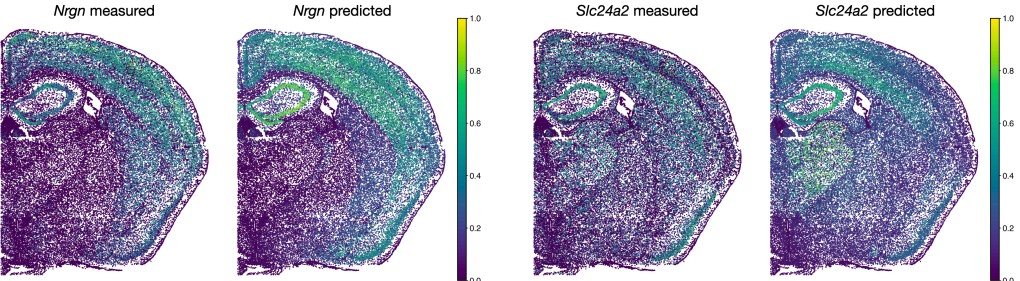

(a) Gene *Nrgn*: measured vs. predicted expression    (b) Gene *Slc24a2*: measured vs. predicted expression

Figure 1: **Exp. 2**: Spatial visualization of two mouse brain sections, contrasting observed vs. predicted (using ULFGW) spatial distributions of expression levels, for two different genes.

## 4    Experiments

We focus first in **Exp. 1** on demonstrating the empirical benefits of the TI variant of our algorithm to solve linear ULOT, as implemented in Alg. 3 vs. Alg. 5; that algorithm is subsequently used as an inner routine to solve all quadratic ULR problems. We compare in **Exp. 2** *unbalanced* low-rank (ULR) solvers to *balanced* low-rank (LR) counterparts on a spatial transcriptomics task, and follow in **Exp. 3** by comparing ULR solvers to entropic (E) counterparts on a smaller task, to accommodate entropic solvers' quadratic complexity. We conclude in **Exp. 4** by comparing ULR solvers to [Thual et al., 2022], which can learn a sparse transport coupling, in the unbalanced FGW setting.

**Datasets.**    We consider two real-world datasets, described in B.1, and two synthetic datasets, that are large enough to showcase our solvers. The real-world datasets consist of both a shared feature space, used to compute the costs matrices for the linear term in the OT and FGW settings, as well as geometries that are specific to each source $s$ and target $t$ measures, and which are used to compute the costs matrices for the quadratic term in the GW and FGW settings. In **Exp. 1**, we sim-

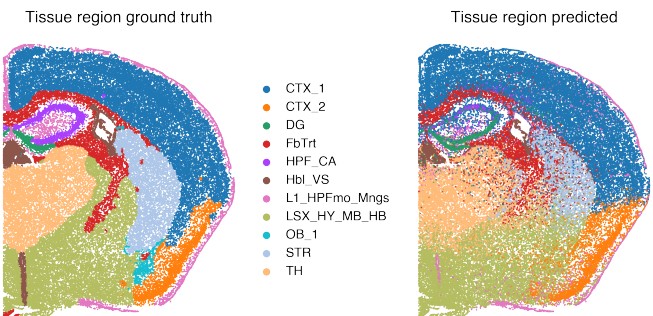

Figure 2: Visualization of measured and predicted tissue regions in the mouse brain in **Exp. 2**

ply consider two isotropic Gaussians in $d = 30$ to evaluate the performance of the TI variant on a liner problem. We use the mouse brain STARmap spatial transcriptomics data from [Shi et al., 2022] for **Exp. 2** and **Exp. 3**. We use data from the Individual Brain Charting dataset [Pinho et al., 2018], to replicate the settings of [Thual et al., 2022], in **Exp. 4**.

**Metrics.**    Following Klein et al. [2023], we evaluate maps by focusing on the two following metrics: (i) **pearson correlation** $\rho$ computed between the (ground truth) source $s$ *feature* matrix $F^s \in \mathbb{R}^{n \times d}$, and the barycentric projection of the target $t$ to the source scaled by the target marginals $b^t$. Writing $P$ as the transport matrix from source to target, this can be computed as $P\mathrm{diag}(\frac{1}{b^t})F^t$; (ii) **F1 score** when assessing class transfer (among 11 possible classes), computed between the original source vector of labels $l^s$, taken in $\{1, \cdots, 11\}^n$, and the inferred labels for the same points, predicted for each $i$ by taking the $\mathrm{argmax}_j B_{i,j}$, where $B$ is a matrix of $n \times 11$ row probabilities, each the barycentric projection of the target $t$ one-hot encoded labels $L^t \in \{0, 1\}^{m \times 11}$, $B := P\mathrm{diag}(\frac{1}{b^t})L^t$.

**Experiment 1: Benchmarking The Translation Invariant Variant.**    We evaluate the effect of the proposed TI procedure on the computational cost of ULR solvers: We compare the time taken when solving unbalanced LR problems, with or without using the TI objective. In Figure 3, we compare the execution time (using our `ott-jax` implementation, and a single NVIDIA GeForce RTX 2080

Ti card) of unbalanced LR Sinkhorn on large and high dimensional Gaussian distributions. The results presented are averaged over 10 random seeds with error bars. We use a $\delta = 10^{-9}$ convergence threshold and 1000 maximal number of iterations for Dykstra, in 64-bit precision. We observe that the use of our proposed TI objective is consistently beneficial when solving ULR problems. See also Appendix B.3 for additional experiments.

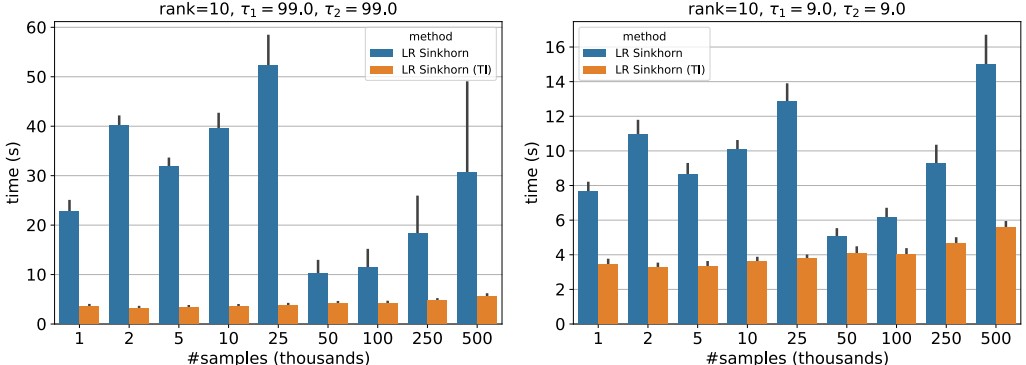

Figure 3: Execution time of unbalanced LR Sinkhorn, with (Alg. 3) or without (Alg. 5) the TI variant. We fix the rank to $r = 10$; $n$ points (displayed in thousands) are sampled from two Gaussian distributions in $d = 30$ of means respectively $-1.2$ and $1.3$, and standard deviations 1 and 0.2. (left) displays large $\tau$ (close to balanced), (right) is smaller $\tau$ (more unbalanced). We use the *same convergence threshold* for the outer loop, for all sample sizes. As $n$ gets bigger, this results in a relatively *looser* threshold, explaining why timings can slightly decrease w.r.t. $n$. What matters is, therefore, the comparative performance of TI vs non-TI for a fixed $n$, *not the behaviour w.r.t. $n$.*

**Experiment 2: ULOT vs. LOT on Gene Expression / Cell Type Annotation.** We evaluate the accuracy of ULOT solvers for a large-scale spatial transcriptomics task, using gene expression mapping and cell type annotation. We compare it to the balanced LR alternative using the Pearson correlation $\rho$ as described in the metrics section. We leverage two coronal sections of the mouse brain profiled by STARmap spatial transcriptomics by

| solver | mass % | val $\rho$ | test $\rho$ | F1 mac. | F1 mic. | F1 weig. |
|--------|--------|-----------|------------|---------|---------|----------|
| LOT    | 1.000  | 0.282     | 0.386      | 0.210   | 0.411   | 0.360    |
| ULOT   | 0.889  | 0.301     | 0.409      | 0.200   | 0.425   | 0.363    |
| LGW    | 1.000  | 0.227     | 0.288      | 0.487   | 0.716   | 0.692    |
| ULGW   | 1.001  | 0.222     | 0.287      | 0.463   | 0.701   | 0.665    |
| LFGW   | 1.000  | 0.365     | 0.443      | 0.576   | 0.720   | 0.714    |
| ULFGW  | 0.443  | **0.379** | **0.463**  | **0.582** | **0.733** | **0.724** |

Table 1: **Exp.2**, Results for spatial transcriptomics dataset (brain coronal section from Shi et al. [2022]).

[Shi et al., 2022]. They consist of $n \approx 40,000$ cells in both the source and target brain section. Each cell is described by 1000 gene features, in addition to 2D spatial coordinates. As a result $A, B$ are $\approx 40k \times 40k$, and the fused term $C$ is a squared-Euclidean distance matrix on 30D PCA space computed on the gene expression space. We selected 10 marker genes for the validation and test sets from the *HPF_CA* cluster. We run an extensive grid search as reported in B.2, we pick the best hyperparameters combination using performance on the 10 validation genes as a criterion, and we report that metric on the other genes in Table 1, as well as qualitative results in Figure 1 and Figure 2. Clearly, ULFGW is the best performing solver across all metrics. Interestingly, the ULOT does not consistently outperforms its balanced version, and unbalancedness seems to hurt performance for the LGW solvers. Nevertheless, both solvers display inconsistent performance across metrics, whereas the ULFGW and LFGW are consistently superior to the rest of the solvers. These results highlight how the flexibility given by the FGW formulation to leverage common and disparate geometries, paired with the unbalancedness relaxation, can provide state of the art algorithms for matching problems in large-scale, real world biological problems.

**Experiment 3: ULOT vs. UEOT.** We compare the performance of ULOT solvers to their unbalanced entropic alternatives (UEOT). We use the same datasets as in **Exp. 2**, but must pick a smaller subset (Olfactory bulb), to avoid OOM errors for entropic UGW solvers, since they cannot handle the $40k$ sizes considered in **Exp. 2** (see B.1). This results in $n \approx 20,000$ source and $\approx 15,000$ target

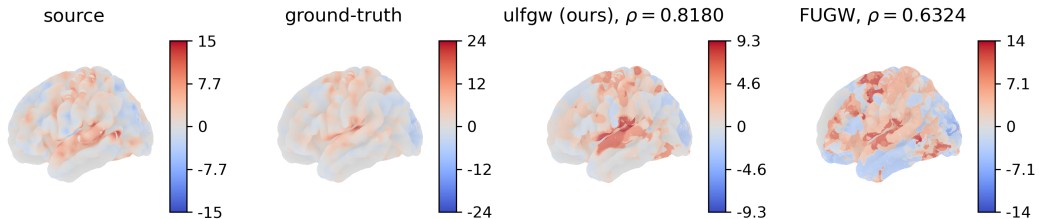

Figure 4: Visualization of measured and predicted *right auditory click* contrast map in **Exp.4**.

cells, and 1000 genes. Similar to **Exp. 2**, the fused term $C$ is a squared-Euclidean distance matrix on 30-D PCA space, computed on gene expressions. As done in **Exp. 2**, we select 10 marker genes

for the validation and 10 genes for the test set, from cluster *OB_1*. We run an extensive grid search, as in **Exp. 2** and B.2. Table 2 shows that ULFGW outperforms entropic solvers w.r.t. Pearson correlation $\rho$, but is worse when considering F1 scores. On the other hand, ULFGW confirms its superiority compared to the balanced alternative LFGW. Taken together, these results suggest that while unbalanced

| solver | mass % | val $\rho$ | test $\rho$ | F1 mac. | F1 mic. | F1 weig. |
|---|---|---|---|---|---|---|
| UEOT | 1.012 | 0.368 | 0.479 | 0.511 | 0.763 | 0.751 |
| LOT | 1.000 | 0.335 | 0.440 | 0.511 | 0.760 | 0.751 |
| ULOT | 0.998 | 0.356 | 0.461 | 0.518 | 0.770 | 0.762 |
| UEFGW | 1.015 | 0.343 | 0.475 | **0.564** | **0.839** | **0.831** |
| LFGW | 1.000 | 0.348 | 0.453 | 0.512 | 0.762 | 0.753 |
| ULFGW | 0.339 | **0.368** | **0.491** | 0.556 | 0.826 | 0.818 |

Table 2: **Exp. 3**: Results for spatial transcriptomics dataset (Olfactory bulb section from Shi et al. [2022]).

LR solvers are on par with unbalanced entropic solvers in terms of performance, in small data regimes, they remain much faster and can unlock the applications of unbalanced OT to larger scales.

**Experiment 4: ULOT to align brain meshes.** In this experiment, we compare the performance of our ULFGW solver to FUGW-sparse [Thual et al., 2022], an alternative approach to solve unbalanced FGW problems, using a two-scale (corse/fine grid) approach to handle large sample sizes. This method was demonstrated to be effective in aligning brain anatomies, encompassing both mesh structures and functional signals associated with each vertex. For their empirical analysis, they use the individual brain charting dataset [Pinho et al., 2018].

In the absence of other information in the original paper, we draw inspiration from Pinho et al.'s smaller scale notebook implementations: We embed the $n \approx 160,000$ vertices of the `fsaverage7` mesh, into a 30-d space, using an approximation of the geodesic distances with landmark multi-dimensional scaling [De Silva and Tenenbaum, 2004] where 2048 points were used as anchors. Each vertex has an associated functional signal that entails 22 features. For both the quadratic and linear terms, we compute the costs based on the squared Euclidean distance. The coarse grid for FUGW-sparse is built using one-

| solver | mass | val $\rho$ | test $\rho$ |
|---|---|---|---|
| FUGW-sparse | 0.999 | 0.492 | 0.472 |
| LFGW | 1.000 | 0.513 | **0.663** |
| ULFGW | 0.981 | **0.533** | 0.643 |

Table 3: Results on the brain anatomy with functional signal data from Pinho et al. [2018] in **Exp.4**.

tenth of $n$, i.e. $\approx 16k$ points. We evaluate all methods by comparing the performance of the best hyperparameter combination, based on the average correlation between the barycentric projection and ground-truth value of 5 features, across a test set of 5 contrast maps. In Table 3, we observe that ULFGW and LFGW outperform FUGW-sparse. In this setting, there is no clear evidence that the unbalanced version performs better than its balanced counterpart for low-rank methods. See also Appendix B.2 for additional experimental details and results.

**Conclusion.** The practical success of OT methods to natural sciences demonstrates the relevance of OT to their analysis pipelines. Practitioners must, however, often deal with the poor scalability of OT algorithms, as well as their rigid assumptions w.r.t. mass conservation. While Low-rank OT approaches hold the promise of scaling OT methods to large sizes, unbalanced formulations have proved useful to relax mass conservation for entropic OT solvers. We have proposed in this paper to merge these two strains, and demonstrated the practical relevance of these unbalanced low-rank solvers on various challenging alignment tasks.

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
