$$\nabla_g \mathcal{H}_k := \alpha \left( \langle C, P_k \rangle \mathbf{1}_r - |g_k| \omega_k^{\mathrm{lin}}/g_k^2 \right) - 4\bar{\alpha} \omega_k^{\mathrm{quad}}/g_k^2$$

$$[\omega_k^{\mathrm{lin}}]_i := [Q_k^T CR_k]_{i,i}, \ [\omega_k^{\mathrm{quad}}]_i := [Q_k^T AP_k BR_k]_{i,i} \ \forall i \in \{1, \dots, r\}.$$

These steps are summarized in Algorithm 6, proposed in the appendix. These steps result usually in a quadratic complexity, both in time and memory, with respect to the number of points $n$ and $m$. These complexities become linear as soon as all three matrices $C, A, B$ admit a low-rank factorization.

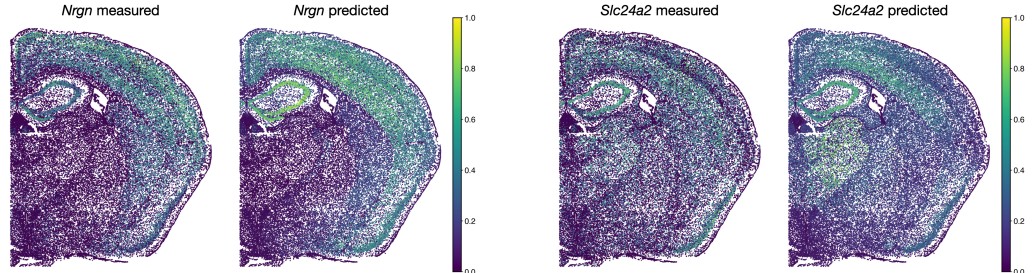

(a) Visualization of measured and predicted gene expression of *Nrgn*.

(b) Visualization of measured and predicted gene expression of *Slc24a2*.

Figure 1: Spatial visualization of the two mouse brain sections used in **Exp. 2**

# 4 Experiments

We focus first in **Exp. 1** on demonstrating the empirical benefits of the translation invariant (TI) variant of our algorithms, as implemented in Algorithm 3, and which is subsequently used as an inner routine to solve ULR problems. We compare in **Exp. 2** *unbalanced* low-rank (ULR) solvers to *balanced* low-rank (LR) counterparts, and follow in **Exp. 3** by comparing ULR solvers to entropic (E) counterparts. We conclude in **Exp. 4** by comparing ULR solvers to [Thual et al., 2022], which can learn a sparse transport coupling, in the unbalanced FGW setting.

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

**Experiment 4: ULOT to align brain meshes.** In this experiment, we compare the performance of our ULFGW solver to FUGW-sparse, a new approach of the unbalanced FGW problem based on a full-rank formulation proposed in Thual et al. [2022]. This method was demonstrated to be effective in aligning brain anatomies, encompassing both mesh structures and functional signals associated with each vertex. For their empirical analysis, they utilized the Individual Brain Charting dataset Pinho et al. [2018].

The dataset uses the *fsaverage7* mesh, which describes $n \approx 160,000$ vertices. We embed them into a 30-dimensional embedding space using an approximation of the geodesic distances with landmark multi-dimensional scaling [De Silva and Tenenbaum, 2004] where 2048 points were used as anchors. Each vertex has an associated functional signal that entails 22 features. For both the quadratic and linear terms, we compute the costs based on the squared Euclidean distance. We evaluate the performance of the method by comparing each best hyperparameter combinations based on the average correlation

| solver | mass | val $\rho$ | test $\rho$ |
|---|---|---|---|
| FUGW-sparse | 0.999 | 0.492 | 0.472 |
| LFGW | 1.000 | 0.513 | **0.663** |
| ULFGW | 0.981 | **0.533** | 0.643 |

Table 3: Results on the brain anatomy with functional signal data from Pinho et al. [2018] in **Exp.4**.

between the barycentric projection and ground-truth value of 5 features, across a test set of 5 contrast maps. See also Appendix B.2 for additional experimental details and results. In Table 3, we observe that ULFGW and LFGW outperforms FUGW-sparse. In this setting, there is no clear evidence that the unbalanced version performs better than its balanced counterpart for low-rank methods.

**Conclusion.** Recent practical successes of OT methods to natural sciences have demonstrated the relevance of OT to their analysis pipelines, but have also shown, repeatedly, that a certain degree of freedom to depart from the rigid assumption of mass conservation is needed in practice. On the other hand, and across the same range of applications, low-rank approaches can hold the promise of scaling OT methods to relevant sample sizes for natural sciences. This paper merges these two strains and demonstrate the practical relevance of these novel algorithms.

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

# Appendix

## A  Algorithms

---
**Algorithm 5** ULR-Dykstra$(a, b, \boldsymbol{\xi}, \gamma, \tau_1, \tau_2, \delta, \alpha)$

---
**Inputs:** $a, b, \boldsymbol{\xi} = (\xi^{(1)}, \xi^{(2)}, \xi^{(3)}), \gamma, \tau_1, \tau_2, \delta$
$v_1 = v_2 = \mathbf{1}_r, u_1 = \mathbf{1}_n, u_2 = \mathbf{1}_m$
**repeat**
$\quad \tilde{v}_1 = v_1, \ \tilde{v}_2 = v_2, \tilde{u}_1 = u_1, \tilde{u}_2 = u_2$
$\quad u_1 = \left(\frac{a}{\xi^{(1)}v_1}\right)^{\frac{\tau_1}{\tau_1 + 1/\gamma}}, \quad u_2 = \left(\frac{b}{\xi^{(2)}v_2}\right)^{\frac{\tau_2}{\tau_2 + 1/\gamma}},$
$\quad g = \left(\xi^{(3)} \odot (\xi^{(1)})^T u_1 \odot (\xi^{(2)})^T u_2\right)^{1/3}, \ v_1 = \frac{g}{(\xi^{(1)})^T u_1}, \ v_2 = \frac{g}{(\xi^{(2)})^T u_2}$
**until** $\frac{1}{\gamma} \max(\|\log(u_i/\tilde{u}_i)\|_\infty, \|\log(v_i/\tilde{v}_i)\|_\infty) < \delta;$
**Result:** $\text{diag}(u_1)\xi_k^{(1)} \text{diag}(v_1), \ \ \text{diag}(u_2)\xi_k^{(2)} \text{diag}(v_2), \ \ g$

---

---
**Algorithm 6** ULFGW$(A, B, a, b, r, \gamma_0, \tau_1, \tau_2, \delta)$

---
**Inputs:** $A, B, C, a, b, r, t, \gamma_0, \tau_1, \tau_2, \delta, \alpha$
$Q, R, g \leftarrow$ Initialization as proposed in [Scetbon and Cuturi, 2022]
**repeat**
$\quad \tilde{Q} = Q, \ \tilde{R} = R, \ \tilde{g} = g,$
$\quad \nabla_Q = \alpha|g|CR \text{diag}(1/g) + \bar{\alpha}\left(2A^{\odot 2}Q\mathbf{1}_r\mathbf{1}_r^T + 4AQ\text{diag}(1/g)R^T BR\text{diag}(1/g)\right),$
$\quad \nabla_R = \alpha|g|C^T Q \text{diag}(1/g) + \bar{\alpha}\left(2B^{\odot 2}R\mathbf{1}_r\mathbf{1}_r^T + 4BR\text{diag}(1/g)Q^T AQ\text{diag}(1/g)\right),$
$\quad \omega^{\text{lin}} \leftarrow \mathcal{D}(Q^T CR), \ \ \omega^{\text{quad}} \leftarrow \mathcal{D}(Q^T AQ\text{diag}(1/g)R^T BR)$
$\quad \nabla_g = \alpha\left(\langle C, Q\text{diag}(1/g)R^T\rangle\mathbf{1}_r - |g_k|\omega^{\text{lin}}/g^2\right) - 4\bar{\alpha}\omega^{\text{quad}}/g^2,$
$\quad \gamma \leftarrow \gamma_0/\max(\|\nabla_Q\|_\infty^2, \|\nabla_R\|_\infty^2, \|\nabla_g\|_\infty^2),$
$\quad \xi^{(1)} \leftarrow Q \odot \exp(-\gamma\nabla_Q), \ \xi^{(2)} \leftarrow R \odot \exp(-\gamma\nabla_R), \ \xi^{(3)} \leftarrow g \odot \exp(-\gamma_k\nabla_g),$
$\quad Q, R, g \leftarrow$ ULR-TI-Dykstra$(a, b, \boldsymbol{\xi}, \gamma, \tau_1, \tau_2, \delta)$ (Alg. 3)
**until** $\Delta((Q, R, g), (\tilde{Q}, \tilde{R}, \tilde{g}), \gamma) < \delta;$
**Result:** $Q, R, g$

---

## B  Experiments

### B.1  Datasets and preprocessing

We downloaded the two publicly available datasets from the respective publications:

- STARmap mouse brain sections from [Shi et al., 2022]
- Brain mesh anatomy and functional signal from [Pinho et al., 2018]

We reprocessed the datasets using standard tools from the SCANPY pipeline [Wolf et al., 2018]. Specifically, we log-normalized gene expression of all genes present in dataset. We selected two brain coronal sections for **Exp.1** and two Coronal Olfactory Bulb (OB) sections for **Exp.2**, from the STARmap dataset. For **Exp.3**, we used the meshes together with their functional signal of the brains to recapitulate **Exp.1** in [Thual et al., 2022]. A visualization of the STARmap dataset for the two subsets used in **Exp.1** and **Exp.2** can be seen in Figure 5 and an overview of the cell type proportions present in each of the section pairs can be see in Figure 6. These visualization highlight the differences in terms of spatial organization and cell type proportions of the brain sections used in the experiment.

### B.2  Experimental settings

For **FUGW-sparse** presented in Table 3, we compute the coupling in 2 stages: (i) similarly as in Thual et al. [2022], we subsample the mesh to 10% of the points using Ward's algorithm and compute

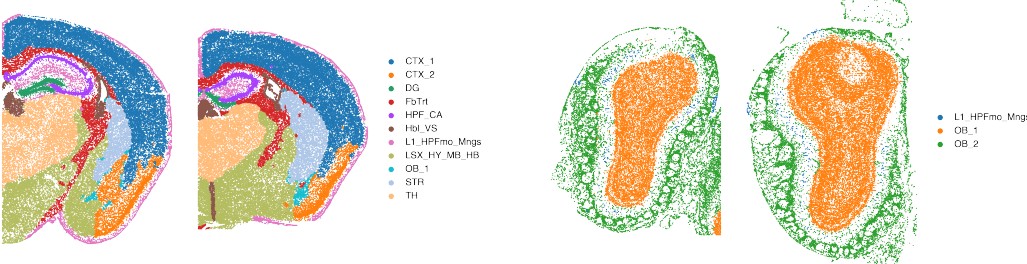

(a) Visualization of the brain coronal sections used in **Exp.1**.

(b) Visualization of the OB sections used in **Exp.2**.

Figure 5: Spatial visualization of the two mouse brain sections used in **Exp.1**.

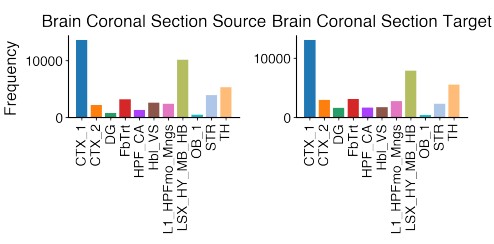 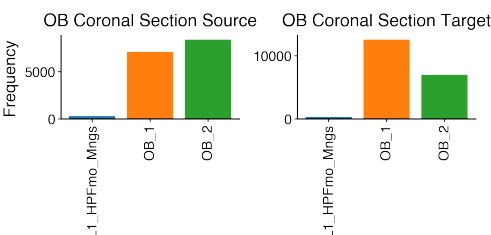

(a) Visualization of cell type frequencies for the brain coronal sections used in **Exp.1**.

(b) Visualization of cell type frequencies for the OB sections used in **Exp.2**.

Figure 6: Cell type frequencies of the datasets used in **Exp.1** and **Exp.2**.

the coarse optimal transport coupling. And (ii) we then use this coarse coupling to define a sparsity mask on the full mesh by selecting for each source (target) vertex the most coupled target (source) vertex and its neighbors within $\frac{4}{max\_distance}$ radius using the approximation of the geodesic distances. This mask is then used to compute the fine-grained sparse coupling.

For all experiments, we ran the grid search as defined by 4 and selected the best set of hyperparameters based on the validation correlation. We report results of top performing hyperparameters for the evaluated algorithms in Table 5 for **Exp.1**, Table 6 for **Exp.2** and Table 7 for **Exp.3**

|  | values |
|---|---|
| **rank** | 10, 50, 100 |
| **reg (ours)** | 0.0, 0.001, 0.01 |
| **reg (fugw-sparse)** | 0.0001, 0.001, 0.01 |
| **tau1** | 0.1, 1.0, 100.0 |
| **tau2** | 0.1, 1.0, 100.0 |

Table 4: Hyperparameters considered in our grid-search.

| solver | rank | tau1 | tau2 | temp | reg | mass | val $\rho$ | test $\rho$ | F1-mac | F1-mic | F1-wei |
|---|---|---|---|---|---|---|---|---|---|---|---|
| lot | 10 | - | - | 0.200 | 0.010 | 1.000 | 0.282 | 0.386 | 0.210 | 0.411 | 0.360 |
| ulot | 10 | 1.000 | 1.000 | 0.200 | 0.010 | 0.889 | 0.301 | 0.409 | 0.200 | 0.425 | 0.363 |
| lgw | 100 | - | - | 0.200 | 0.001 | 1.000 | 0.227 | 0.288 | 0.487 | 0.716 | 0.692 |
| ulgw | 100 | 100.000 | 100.000 | 0.200 | 0.010 | 1.001 | 0.222 | 0.287 | 0.463 | 0.701 | 0.665 |
| lfgw | 50 | - | - | 0.400 | 0.010 | 1.000 | 0.365 | 0.443 | 0.576 | 0.720 | 0.714 |
| ulfgw | 100 | 0.100 | 0.100 | 0.400 | 0.001 | 0.443 | 0.379 | 0.463 | 0.582 | 0.733 | 0.724 |

Table 5: Results on the large spatial transcriptomics dataset (brain coronal section from [Shi et al., 2022]).

| solver | rank | tau1 | tau2 | temp | reg | mass | val $\rho$ | test $\rho$ | F1-mac | F1-mic | F1-wei |
|--------|------|------|------|------|-----|------|-----------|------------|--------|--------|--------|
| uot | - | 0.909 | 0.999 | 0.400 | 0.100 | 1.012 | 0.368 | 0.479 | 0.511 | 0.763 | 0.751 |
| lot | 10 | - | - | 0.200 | 0.010 | 1.000 | 0.335 | 0.440 | 0.511 | 0.760 | 0.751 |
| ulot | 10 | 1.000 | 100.000 | 0.200 | 0.010 | 0.998 | 0.356 | 0.461 | 0.518 | 0.770 | 0.762 |
| ufgw | - | 0.500 | 0.999 | 0.600 | 0.100 | 1.015 | 0.343 | 0.475 | 0.564 | 0.839 | 0.831 |
| lfgw | 10 | - | - | 0.600 | 0.010 | 1.000 | 0.348 | 0.453 | 0.512 | 0.762 | 0.753 |
| ulfgw | 10 | 0.100 | 0.100 | 0.600 | 0.001 | 0.339 | 0.368 | 0.491 | 0.556 | 0.826 | 0.818 |

Table 6: Results on the small subset STARmap dataset (OB section from [Shi et al., 2022]).

| solver | rank | tau1 | tau2 | reg | reg | mass | val $\rho$ | test $\rho$ |
|--------|------|------|------|-----|-----|------|-----------|------------|
| **fugw-sparse** | - | 1.000 | 0.100 | 0.200 | 0.01 | 0.999 | 0.492 | 0.472 |
| **lfgw** | 100 | - | - | 0.600 | 0.000 | 1.000 | 0.513 | 0.663 |
| **ulfgw** | 100 | 1.000 | 0.100 | 0.600 | 0.001 | 0.981 | 0.533 | 0.643 |

Table 7: Results on the brain anatomy and functional signal from [Pinho et al., 2018]).

| experiment | | val $\rho$ | tst $\rho$ | F1-mac | F1-mic | F1-wei |
|------------|------|-----------|-----------|--------|--------|--------|
| **Exp. 1** | mean | 0.362 | 0.449 | 0.546 | 0.687 | 0.677 |
| | std | 0.027 | 0.022 | 0.054 | 0.062 | 0.061 |
| **Exp. 2** | mean | 0.356 | 0.463 | 0.538 | 0.800 | 0.791 |
| | std | 0.008 | 0.018 | 0.021 | 0.031 | 0.032 |

Table 8: Effect of k-means initialization [Scetbon and Cuturi, 2022]. We report mean and standard deviation of *test* criterion for ULFGW, with the best hyperparameter on validation data for each experiment. We use 5 initial seeds for **Exp. 1**. We observe more variability in validation performance for **Exp. 2**, and therefore start with 10 seeds, pruning the lowest performing 5 seeds.

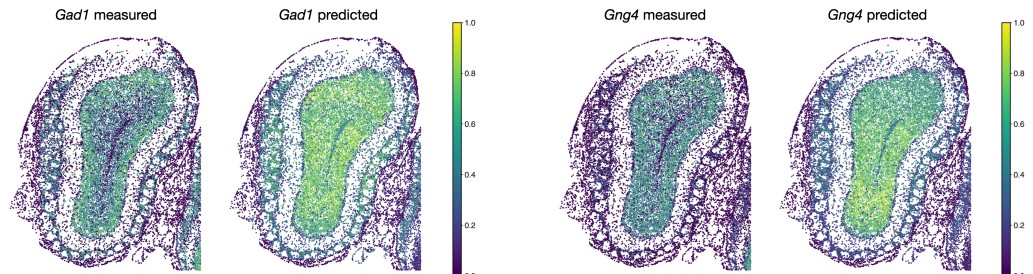

(a) Visualization of measured and predicted gene expression of *Gad1*.

(b) Visualization of measured and predicted gene expression of *Gnrg4*.

Figure 7: Measured and predicted gene expression for the small subset STARmap dataset (OB section from [Shi et al., 2022]) for ULRFGW.

## B.3 Additional Experiments on the TI procedure

Here, we provide additional experiments in order to measure the effect of the TI version on the computational performance of LR solvers.

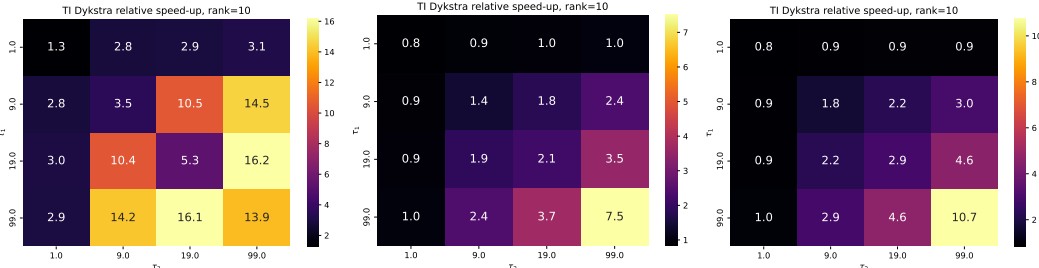

Figure 8: Speed-up of TI variant when varying $\tau_1, \tau_2$. *(left)* ULR **Sinkhorn** for $n = 25k$ points in 30d, rank=10, as in Figure 3. *(middle)* ULR-**GW** for $n = 50k$ points in src/tgt in 30d / 40d, means -1.2 / 1.3, std 1/0.2 between Gaussians.*(right)* ULR-**GW** as in middle, but data comes from **GMMs** (sklearn's blobs) with 10/15 clusters.

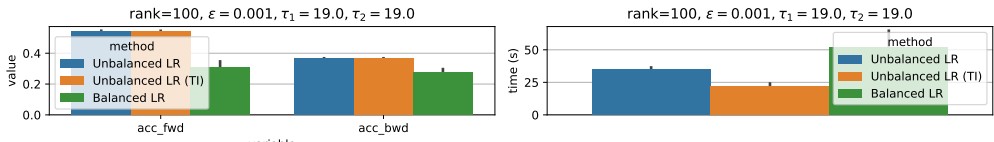

Figure 9: We used the Lobby room from Stanford 3D Indoor Scene Dataset (S3DIS) [Armeni et al., 2016] that consists of 1M points. *(left)* source-to-target, and target-to-source accuracies on the scene data, in a pure GW setting of balanced, unbalanced and unbalanced (TI variant). We use random initializer for all, 150 max outer iterations. The backward accuracy is comparable to what is mentioned in `https://2021.ecmlpkdd.org/wp-content/uploads/2021/07/sub_ 949.pdf` ($\approx 0.41$). *(right)* same as left, but showing timings, also comparable to those mentioned in the Quantized GW paper [Chowdhury et al., 2021] (10 min.)

## C Proofs

### C.1 Proof of Proposition 1

Let $n, m \geq r \geq 1$, $\gamma > 0$, , $\boldsymbol{\xi} := (\xi^{(1)}, \xi^{(2)}, \xi^{(3)})$ where $\xi^{(1)} \in \mathbb{R}_+^{n \times r}$, $\xi^{(2)} \in \mathbb{R}_+^{m \times r}$ and $\xi^{(3)} \in \mathbb{R}_+^r$ and let us recall that $\mathrm{KL}(\cdot, \cdot)$ is the generalized Kullback-Leibler divergence defined as $\mathrm{KL}(p|q) := \sum_i p_i \log(p_i/q_i) + q_i - p_i$. Then observe that

$$\min_{(Q,R,g) \in \Pi_r} \frac{1}{\gamma} \left[ \mathrm{KL}(Q, \xi^{(1)}) + \mathrm{KL}(R, \xi^{(2)}) + \mathrm{KL}(g, \xi^{(3)}) \right] + \tau_1 \mathrm{KL}(Q\mathbf{1}_r|a) + \tau_2 \mathrm{KL}(R\mathbf{1}_r|b)$$

is a convex problem satisfying the Slater's condition and therefore strong duality holds. Therefore we have:

$$\min_{(Q,R,g) \in \Pi_r} \frac{1}{\gamma} \left[ \mathrm{KL}(Q, \xi^{(1)}) + \mathrm{KL}(R, \xi^{(2)}) + \mathrm{KL}(g, \xi^{(3)}) \right] + \tau_1 \mathrm{KL}(Q\mathbf{1}_r|a) + \tau_2 \mathrm{KL}(R\mathbf{1}_r|b)$$

$$= \sup_{\lambda_1, \lambda_2} \min_{Q,R,g} \langle \lambda_1, g - Q^\top \mathbf{1}_n \rangle + \langle \lambda_2, g - R^\top \mathbf{1}_m \rangle + \frac{1}{\gamma} \left[ \mathrm{KL}(Q, \xi^{(1)}) + \mathrm{KL}(R, \xi^{(2)}) + \mathrm{KL}(g, \xi^{(3)}) \right]$$

$$+ \tau_1 \mathrm{KL}(Q\mathbf{1}_r|a) + \tau_2 \mathrm{KL}(R\mathbf{1}_r|b)$$

$$= \sup_{\lambda_1, \lambda_2} \min_Q \frac{1}{\gamma} \mathrm{KL}(Q, \xi^{(1)}) + \tau_1 \mathrm{KL}(Q\mathbf{1}_r|a) + \langle \lambda_1, -Q^\top \mathbf{1}_n \rangle$$

$$+ \min_R \frac{1}{\gamma} \mathrm{KL}(R, \xi^{(2)}) + \tau_2 \mathrm{KL}(R\mathbf{1}_r|b) + \langle -\lambda_2, R^\top \mathbf{1}_m \rangle + \min_g \frac{1}{\gamma} \mathrm{KL}(g, \xi^{(3)}) + \langle g, \lambda_1 + \lambda_2 \rangle.$$

Now consider

$$\min_g \frac{1}{\gamma} \mathrm{KL}(g, \xi^{(3)}) + \langle g, \lambda_1 + \lambda_2 \rangle$$

and observe that this problem can be solved explicitly. The first-order optimality condition gives us that $g^* = \exp(-\gamma(\lambda_1 + \lambda_2)) \odot \xi^{(3)}$ solves the problem and

$$\min_g \frac{1}{\gamma} \mathrm{KL}(g, \xi^{(3)}) + \langle g, \lambda_1 + \lambda_2 \rangle = -\frac{1}{\gamma} \langle \exp(-\gamma(\lambda_1 + \lambda_2)), \xi^{(3)} \rangle + \langle \xi^{(3)}, \mathbf{1} \rangle.$$

Let us now focus on the following convex optimization problem,

$$\min_Q \frac{1}{\gamma}\mathrm{KL}(Q,\xi^{(1)}) + \tau_1\mathrm{KL}(Q\mathbf{1}_r|a) + \langle -\lambda_1, Q^\top\mathbf{1}_n\rangle \tag{18}$$

and note that it admits a unique solution due to the strict convexity of $Q \to \mathrm{KL}(Q,\xi^{(1)})$. Then by denoting $F_{\tau,z}(s) := \tau\mathrm{KL}(s|z)$ and $G_\lambda(s) := \langle s, -\lambda\rangle$, and by applying the Fenchel-Rockafellar theorem [Rockafellar, 1970], we obtain that strong duality holds, the dual problem of (18) is

$$\sup_{f_1,h_1} -F^*_{\tau_1,a}(-f_1) - G^*_{\lambda_1}(-h_1) - \frac{1}{\gamma}\langle\exp(\gamma(f_1+h_1)),\xi^{(1)}\rangle$$

and that $(f_1,h_1)$ solves the dual if and only if $-f_1 \in \partial F_{\tau_1,a}(Q\mathbf{1}_r), -h_1 \in \partial G_{\lambda_1}(Q^\top\mathbf{1}_n)$ and $Q = \mathrm{diag}(\exp(\gamma f_1))\xi^{(1)}\mathrm{diag}(\exp(\gamma h_1))$ where $Q$ is the solution of (18). Recall that here we denote for any convex set $X \in \mathbb{R}^q$ and function $f : X \to \mathbb{R}\cup\{+\infty\}$, $f^*$ its convex conjugate defined for any $y \in X^* := \{x^*$ s.t. $\sup_{x\in X}\langle x, x^*\rangle - f(x) < +\infty\}$ by $f^*(y) := \sup_{x\in X}\langle x, y\rangle - f(x)$ and $\partial f(x) := \{y$ s.t. $f(x') - f(x) \geq \langle y, x-x'\rangle \ \forall x' \in X\}$. Now remarks that

$$G^*_{\lambda_1}(-h_1) = \sup_s\langle s, \lambda_1 - h_1\rangle = \left\{\begin{array}{ll} +\infty & \text{if } \lambda_1 \neq h_1 \\ 0 & \text{otherwise} \end{array}\right. .$$

therefore $G^*_{\lambda_1}$ ensures that $\lambda_1 = h_1$. Similarly we obtain that

$$\min_R \frac{1}{\gamma}\mathrm{KL}(R,\xi^{(2)}) + \tau_2\mathrm{KL}(r\mathbf{1}_r|b) + \langle -\lambda_2, R^\top\mathbf{1}_m\rangle \tag{19}$$

is equal to its dual defined as

$$\sup_{f_2,h_2} -F^*_{\tau_2,b}(-f_2) - G^*_{\lambda_2}(-h_2) - \frac{1}{\gamma}\langle\exp(\gamma(f_2+h_2)),\xi^{(2)}\rangle$$

where again

$$G^*_{\lambda_2}(-h_2) = \left\{\begin{array}{ll} +\infty & \text{if } \lambda_2 \neq h_2 \\ 0 & \text{otherwise} \end{array}\right.$$

and with the primal-dual relationship $R = \mathrm{diag}(\exp(\gamma f_2))\xi^{(2)}\mathrm{diag}(\exp(\gamma h_2))$ such that $-f_2 \in \partial F_{\tau_2,b}(R\mathbf{1}_r), -h_2 \in \partial G_{\lambda_2}(R^\top\mathbf{1}_m)$. Finally the dual can be written as

$$\sup_{\lambda_1,\lambda_2}\sup_{f_1,h_1} -F^*_{\tau_1,a}(-f_1) - G^*_{\lambda_1}(-h_1) - \frac{1}{\gamma}\langle\exp(\gamma(f_1+h_1)),\xi^{(1)}\rangle$$

$$+ \sup_{f_2,h_2} -F^*_{\tau_2,b}(-f_2) - G^*_{\lambda_2}(-h_2) - \frac{1}{\gamma}\langle\exp(\gamma(f_2+h_2)),\xi^{(2)}\rangle$$

$$- \frac{1}{\gamma}\langle\exp(-\gamma(\lambda_1+\lambda_2)),\xi^{(3)}\rangle + \langle\xi^{(3)},\mathbf{1}\rangle$$

and using the definition of $G^*_{\lambda_1}(-h_1)$ and $G^*_{\lambda_2}(-h_2)$, we obtain the desired dual up to an additive constant ($\langle\xi^{(3)},\mathbf{1}\rangle$) which does not affect the solution of the problem and conclude the proof.

### C.2 On the Iterations of the Dykstra's Algorithm

Recall that we propose to consider an alternate maximization scheme to solve (8). Starting from $h_1^{(0)} = h_2^{(0)} = \mathbf{0}_r$, we apply for $\ell \geq 0$ the following updates (dropping iteration number $k$ in (7) for simplicity):

$$f_1^{(\ell+1)} := \arg\sup_z \mathcal{D}(z, h_1^{(\ell)}, f_2^{(\ell)}, h_2^{(\ell)}), \ f_2^{(\ell+1)} := \arg\sup_z \mathcal{D}(f_1^{(\ell+1)}, h_1^{(\ell)}, z, h_2^{(\ell)}),$$

$$(h_1^{(\ell+1)}, h_2^{(\ell+1)}) := \arg\sup_{z_1,z_2} \mathcal{D}(f_1^{(\ell+1)}, z_1, f_2^{(\ell+1)}, z_2).$$

where

$$\mathcal{D}(f_1, h_1, f_2, h_2) = -F^*_{\tau_1,a}(-f_1) - \frac{1}{\gamma}\langle e^{\gamma(f_1\oplus h_1)} - 1, \xi^{(1)}\rangle - F^*_{\tau_2,b}(-f_2) - \frac{1}{\gamma}\langle e^{\gamma(f_2\oplus h_2)} - 1, \xi^{(2)}\rangle$$

$$- \frac{1}{\gamma}\langle e^{-\gamma(h_1+h_2)} - 1, \xi^{(3)}\rangle.$$

Let us consider the first update of the scheme that consists in solving

$$f_1^{(\ell+1)} := \arg\sup_z \mathcal{D}(z, h_1^{(\ell)}, f_2^{(\ell)}, h_2^{(\ell)})$$

To solve this problem, we again apply the Fenchel-Rockafellar theorem [Rockafellar, 1970] and obtain that

$$\sup_{f_1} -F^*_{\tau_1,a}(-f_1) - \frac{1}{\gamma}\langle \exp(\gamma(f_1 + h_1)), \xi^{(1)}\rangle = \min_s F_{\tau_1,a}(s) + \frac{1}{\gamma}\mathrm{KL}(s|\xi^{(1)}\exp(\gamma h_1))$$

and the optimality condition gives that $f_1^*$ is solution of the LHS if and only if $s^*$ solves the RHS and belongs to the subdifferential of $f_1 \to \exp(\gamma(f_1+h_1)), \xi^{(1)}\rangle$ at $f_1^*$, that is $s^* = \exp(\gamma f_1^*) \odot \xi^{(1)}\exp(\gamma h_1)$. However the RHS problem can can be solved exactly and one obtained that $s^* = a^{(\tau_1/(1/\gamma+\tau_1))} \odot \xi^{(1)}\exp(\gamma h_1)^{(1/(1/1+\gamma\tau_1))}$, therefore when combined with the previous equation on $s^*$ we obtain that

$$\exp(\gamma f_1^*) = \frac{s^*}{\xi^{(1)}\exp(\gamma h_1)} = \left(\frac{a}{\xi^{(1)}\exp(\gamma h_1)}\right)^{\frac{\tau_1}{1/\gamma+\tau_1}},$$

Similarly, the solution of $\arg\sup_z \mathcal{D}(f_1, h_1, z, h_2)$ is

$$\exp(\gamma f_2^*) = \left(\frac{b}{\xi^{(2)}\exp(\gamma h_2)}\right)^{\frac{\tau_2}{1/\gamma+\tau_2}}.$$

Let us now consider the following optimization problem corresponding to the last update if the alternate maximization scheme, that is

$$(h_1^{(\ell+1)}, h_2^{(\ell+1)}) := \arg\sup_{z_1,z_2} \mathcal{D}(f_1^{(\ell+1)}, z_1, f_2^{(\ell+1)}, z_2).$$

In fact this problem can be solved directly using simply the first-order condition of optimality that gives the two following equations:

$$\exp(\gamma h_1) \odot (\xi^{(1)})^\top \exp(\gamma f_1) - \exp(-\gamma h_1) \odot (\xi^{(3)}) \odot \exp(-\gamma h_2) = 0 \quad \text{and}$$
$$\exp(\gamma h_2) \odot (\xi^{(2)})^\top \exp(\gamma f_2) - \exp(-\gamma h_2) \odot (\xi^{(3)}) \odot \exp(-\gamma h_1) = 0$$

leading to

$$g = (\xi^{3)} \odot (\xi^{(1)})^\top \exp(\gamma f_1) \odot (\xi^{(2)})^\top \exp(\gamma f_2))^{1/3}$$

and

$$\exp(\gamma h_1) = \frac{g}{(\xi^{(1)})^\top \exp(\gamma f_1)}, \quad \exp(\gamma h_2) = \frac{g}{(\xi^{(2)})^\top \exp(\gamma f_2)}.$$

### C.3 Proof of Proposition 2

Let us consider the following optimization problem

$$\mathcal{D}_{\mathrm{TI}}(\tilde{f}_1, \tilde{h}_1, \tilde{f}_2, \tilde{h}_2) := \sup_{\lambda_1,\lambda_2 \in \mathbb{R}} \mathcal{D}(\tilde{f}_1 + \lambda_1, \tilde{h}_1 - \lambda_1, \tilde{f}_2 + \lambda_2, \tilde{h}_2 - \lambda_2)$$

Therefore we have

$$\sup_{\lambda_1,\lambda_2 \in \mathbb{R}} \mathcal{D}(\tilde{f}_1 + \lambda_1, \tilde{h}_1 - \lambda_1, \tilde{f}_2 + \lambda_2, \tilde{h}_2 - \lambda_2)$$
$$= -F^*_{\tau_1,a}(-(\tilde{f}_1 + \lambda_1)) - F^*_{\tau_2,b}(-(\tilde{f}_2 + \lambda_2)) - \frac{1}{\gamma}\langle e^{-\gamma(\tilde{h}_1+\tilde{h}_2)} \odot e^{\gamma(\lambda_1+\lambda_2)}, \xi^{(3)}\rangle + C$$

where $C$ does not depends on $\lambda_1$ and $\lambda_2$. Now observe that

$$F^*_{\tau_1,a}(s) = \sup_x \langle x, s\rangle - \tau_1 \mathrm{KL}(s|a)$$

and by applying the first-order optimality condition, we obtain that $x^* = \exp(s/\tau_1) \odot a$ solves the above optimization problem and

$$F_{\tau_1,a}^*(s) = \tau_1 \langle \exp(s/\tau_1), a \rangle.$$

Similarly,

$$F_{\tau_2,b}^*(s) = \tau_2 \langle \exp(s/\tau_2), b \rangle,$$

Then by appling the first-order optimality condition we obtain the two following equations

$$\exp(-\lambda_1/\tau_1)\langle \exp(-\tilde{f}_1/\tau_1), a \rangle - \exp(\gamma\lambda_1)\langle \exp(\gamma\lambda_2), \xi^{(3)} \odot \exp(-\gamma(\tilde{h}_1 + \tilde{h}_2)) \rangle = 0 \quad \text{and}$$

$$\exp(-\lambda_2/\tau_2)\langle \exp(-\tilde{f}_2/\tau_2), b \rangle - \exp(\gamma\lambda_2)\langle \exp(\gamma\lambda_1), \xi^{(3)} \odot \exp(-\gamma(\tilde{h}_1 + \tilde{h}_2)) \rangle = 0.$$

which is equivalent to

$$\exp\left(\lambda_1 \frac{1/\gamma + \tau_1}{\tau_1/\gamma}\right) = \frac{\langle \exp(-\tilde{f}_1/\tau_1), a \rangle}{\langle \xi^{(3)}, \exp(-\gamma(\tilde{h}_1 + \tilde{h}_2)) \rangle} \exp(-\gamma\lambda_2) \quad \text{and}$$

$$\exp\left(\lambda_2 \frac{1/\gamma + \tau_2}{\tau_2/\gamma}\right) = \frac{\langle \exp(-\tilde{f}_2/\tau_2), b \rangle}{\langle \xi^{(3)}, \exp(-\gamma(\tilde{h}_1 + \tilde{h}_2)) \rangle} \exp(-\gamma\lambda_1)$$

Then applying $\log$ to the system, we obtain that

$$\lambda_1 \gamma \frac{1/\gamma + \tau_1}{\tau_1} = c_1 - \gamma\lambda_2 \quad \text{and}$$

$$\lambda_2 \gamma \frac{1/\gamma + \tau_2}{\tau_2} = c_2 - \gamma\lambda_1$$

where

$$c_1 := \log\left(\frac{\langle \exp(-\tilde{f}_1/\tau_1), a \rangle}{\langle \exp(-\gamma(\tilde{h}_1 + \tilde{h}_2)), \xi^{(3)} \rangle}\right), \quad \text{and} \quad c_2 := \log\left(\frac{\langle \exp(-\tilde{f}_2/\tau_2), a \rangle}{\langle \exp(-\gamma(\tilde{h}_1 + \tilde{h}_2)), \xi^{(3)} \rangle}\right).$$

Finally we obtain a simple linear system and the solution follows.

## C.4 Double Regularizations: Low-rank Structure and Entropy

Our proposed procedure can be easily extended to the case where one wants to add entropy in addition to the low-rank constraint to solve unbalanced low-rank and entropic optimal transport problems. More precisely, let us consider the general case where one aims at solving for any $\varepsilon > 0$

$$\text{ULOT}_{r,\varepsilon}(\mu,\nu) := \min_{(Q,R,g)\in\Pi_r} \underbrace{\langle C, Q \operatorname{diag}(1/g)R^T \rangle}_{\mathcal{L}_C(Q,R,g)} + \underbrace{\tau_1 \text{KL}(Q\mathbf{1}_r|a) + \tau_2 \text{KL}(R\mathbf{1}_r|b) - \varepsilon H(Q,R,g)}_{\mathcal{G}_{a,b,\varepsilon}(Q,R,g)} \tag{20}$$

where $H(Q,R,g) = H(Q) + H(R) + H(g)$ and $H(p) := -\sum_i p_i(\log(p_i) - 1)$. Note that here, compared to (6), we have simply add en entropic term to the objective to smooth the matrices $Q, R$ and the barycenter $g$. To solve this problem, we propose to consider the exact same strategy as the one proposed to solve (6) where we slightly modify $\mathcal{G}_{a,b,\varepsilon}$ and explicitly show the dependency w.r.t. $\varepsilon$. Now by applying the linearzation step of $\mathcal{L}_C(Q,R,g)$, we now aim to solve at iteration $k$ the following optimization problem:

$$(Q_{k+1}, R_{k+1}, g_{k+1}) := \operatorname*{argmin}_{\zeta\in\Pi_r} \frac{1}{\gamma_k}\text{KL}(\zeta,\xi_k) + \varepsilon H(\zeta) + \tau_1\text{KL}(Q\mathbf{1}_r|a) + \tau_2\text{KL}(R\mathbf{1}_r|b) \tag{21}$$

In fact, this problem can be reformulated as a problem of the form (14) where we simply have to modify $\xi_k$ and $\gamma$. Indeed observe that we have

$$\frac{1}{\gamma}\text{KL}(Q|\xi^{(1)}) - \varepsilon H(Q) = \frac{1}{\gamma_\varepsilon}\text{KL}(Q|\xi_\varepsilon^{(1)})$$

where $\gamma_\varepsilon = \frac{1}{1/\gamma+\varepsilon}$ and $\xi_\varepsilon^{(1)} := (\xi^{(1)})^{\gamma_\varepsilon/\gamma}$. Therefore we obtain that

$$\operatorname*{argmin}_{\zeta\in\Pi_r} \frac{1}{\gamma}\text{KL}(\zeta,\xi) + \varepsilon H(\zeta) + \tau_1\text{KL}(Q\mathbf{1}_r|a) + \tau_2\text{KL}(R\mathbf{1}_r|b)$$

$$= \operatorname*{argmin}_{\zeta\in\Pi_r} \frac{1}{\gamma_\varepsilon}\text{KL}(\zeta,\xi_\varepsilon) + \tau_1\text{KL}(Q\mathbf{1}_r|a) + \tau_2\text{KL}(R\mathbf{1}_r|b)$$

where $\boldsymbol{\xi}_\varepsilon := (\xi_\varepsilon^{(1)}, \xi_\varepsilon^{(2)}, \xi_\varepsilon^{(3)})$. Therefore the entropic version of our problem can be solved using the exact same solver as the one proposed in the main paper where only simple updates of the gradient-step $\gamma$ and the kernels $\boldsymbol{\xi}$ are required at each iteration. We summarize the proposed algorithm below.

---

**Algorithm 7** $\mathrm{ULOT}_\varepsilon(C, a, b, r, \gamma_0, \tau_1, \tau_2, \delta)$

---

**Inputs:** $C, a, b, \varepsilon, \gamma_0, \tau_1, \tau_2, \delta$
$Q, R, g \leftarrow$ Initialization as proposed in [Scetbon and Cuturi, 2022]
**repeat**
$\quad \tilde{Q} = Q, \ \tilde{R} = R, \ \tilde{g} = g,$
$\quad \nabla_Q = CR \operatorname{diag}(1/g), \ \nabla_R = C^\top Q \operatorname{diag}(1/g),$
$\quad \omega \leftarrow \mathcal{D}(Q^T CR), \ \nabla_g = -\omega/g^2,$
$\quad \gamma \leftarrow \gamma_0 / \max(\|\nabla_Q\|_\infty^2, \|\nabla_R\|_\infty^2, \|\nabla_g\|_\infty^2),$
$\quad \gamma \leftarrow \frac{1}{1/\gamma + \varepsilon}$
$\quad \xi^{(1)} \leftarrow Q \odot \exp(-\gamma \nabla_Q), \ \xi^{(2)} \leftarrow R \odot \exp(-\gamma \nabla_R), \ \xi^{(3)} \leftarrow g \odot \exp(-\gamma \nabla_g),$
$\quad \xi^{(1)} \leftarrow (\xi^{(1)})^{\gamma_\varepsilon/\gamma}, \ \xi^{(2)} \leftarrow (\xi^{(2)})^{\gamma_\varepsilon/\gamma}, \ \xi^{(3)} \leftarrow (\xi^{(3)})^{\gamma_\varepsilon/\gamma},$
$\quad Q, R, g \leftarrow \mathrm{ULR\text{-}Dykstra}(a, b, \boldsymbol{\xi}, \gamma, \tau_1, \tau_2, \delta)$ (Alg. 5)
**until** $\Delta((Q, R, g), (\tilde{Q}, \tilde{R}, \tilde{g}), \gamma) < \delta$;
**Result:** $Q, R, g$

---