# OpenReview forum: "Unbalanced Low-rank Optimal Transport Solvers"
_NeurIPS.cc/2023/Conference — NeurIPS 2023 poster_

### Official Review · Reviewer_GuCe · 2023-06-22

**Soundness:** 3 good
**Presentation:** 3 good
**Contribution:** 4 excellent
**Rating:** 7
**Confidence:** 4

**Summary:**

The authors focus on the problem of efficiently compute discrete Optimal Transport (OT) problems, which are known to have a cubic complexity w.r.t. the number of input samples.
They propose to approximate it by assuming that the transport plan is a low-rank matrix and propagate this property in the computations, such that the complexity of matrix products is reduced.
This idea has already been proposed to approximate Balanced OT and Gromov-Wasserstein (GW) problems.
The authors extend this to variants called Unbalanced OT, as well as Unbalanced GW and Fused-Unbalanced GW.
The authors introduce these unbalanced OT variants, they derive in each setting the associated low-rank optimization algorithms using mirror descent.
Then they perform experiments on brain and cell biology data, where unbalanced OT was extremely powerful, to assess the performance of these algorithms in an applied setting.

**Strengths:**

- The contributions might look incremental at first (combining unbalanced OT with low-rank OT), but are actually very thorough.
They do not restrict to this one combination, but consider all recent variants of unbalanced OT (Unbalanced OT, GW and Fused GW) which have been developped in the literature in the last years.
They also leverage one recent optimization idea of 'translation-invariance' which improves the computational efficiency of computing Unbalanced OT problems.
All in all, the authors aggregate several works altogether to propose a set of interesting algorithms and tools for practitioners, especially for the biology field.


**Weaknesses:**

- The low-rank approximation is not sufficiently discussed in this paper.
I mentioned that it is considered to accelerate the computations.
However, one might use this variant because it makes sense to have a prior of low-rank transport plan in their applications, because the data has some structural property which could be leveraged.
To my mind, using a low-rank plan seems contradictory with the property that the optimal plan is a (full rank) permutation in some setting.
Could the authors discuss this in detail ? Why does it make sense to have a low-rank plan ? Which kind of data is relevant with such assumption ? This is unclear to me.

- Some experimental illustrations could be provided on the methodological side of their algorithms, for completeness.
The authors propose some optimization problems and algorithms to compute them.
To my mind, the authors do provide complexity per iteration of their algorithms, but the experimental aspect of measuring the time performance of their algorithms is omitted.
In particular, have you checked experimentally that this 'translation-invariance' variant in Section (3.2) does accelerate the computations in your low-rank setting ?
Could you provide a plot of the time to compute OT problems (standard or Sinkhorn vs. low-rank) as a function of the number of samples ?
Also, the use of unbalanced OT involves two extra parameters $(\tau_1, \tau_2)$ which requires cross-validation.
Could the authors give plots of the performance of their biology task as a function of $(\tau_1, \tau_2)$.
This would provide interesting insights on the interpretation of these parameters, and their impact on learning tasks.


**Questions:**

See my question above.

**Limitations:**

They adressed the societal impact of their work.

---

> ### Author Rebuttal · Authors · 2023-08-07
>
> We thank you for your encouraging review.
>
> > **The low-rank approximation is not sufficiently discussed in this paper. I mentioned that it is considered to accelerate the computations. However, one might use this variant because it makes sense to have a prior of low-rank transport plan in their applications, because the data has some structural property which could be leveraged. To my mind, using a low-rank plan seems contradictory with the property that the optimal plan is a (full rank) permutation in some setting. Could the authors discuss this in detail ? Why does it make sense to have a low-rank plan ? Which kind of data is relevant with such assumption ? This is unclear to me.**
>
> ⧐ Your intuition is indeed correct:
>
> In the general case, a coupling matrix $P^\star$ minimizing the linear objective in Eq. 1 (e.g. that returned by an LP solver) will be (i) sparse (up to $n+m-1$ non-zeros values for a $n\times m$ matrix) and (ii) full rank.
>
> The idea of low-rank constraints and/or entropic regularization is therefore to move *away* from these properties.
> - For the low rank approach, this is akeen to requiring a "cluster" structure when computing transport (i.e. all of the mass must go through $r$ virtual points)
> - For entropic regularization, this forces all sources and targets to transfer mass between each other.
>
> Both can be seen as **inductive biases** when trying to match points in large dimensions, that result in substantial computational and statistical gains. Both result in a maximally diffuse / rank 1 transport coupling $ab^T$ in the limit, when $r=1$ or when entropic regularization is infinite.
>
> These aspects are already well covered in the references provided in lines 45-50, which is why we did not use to much space in the introduction for these reminders. Following your comment (and some space gained my moving some of the more technical aspects to the appendix) we will add the sentences above (abridged).
>
> > **To my mind, the authors do provide complexity per iteration of their algorithms,**
>
> ⧐ We have improved this aspect, and have added compute complexities alongside iteration costs to be more explicit, as was done e.g. in [Scetbon'22]
>
> > **but the experimental aspect of measuring the time performance of their algorithms is omitted. In particular, have you checked experimentally that this 'translation-invariance' variant in Section (3.2) does accelerate the computations in your low-rank setting ?**
>
> ⧐ This is an excellent suggestion. We have now added various experiments that illustrate such speed-ups (see Figures 1 and 2 in attached Pdf). We agree this strengthens our submission and will continue exploring to what extent this TI modification is always a "win" (we have seen it become slightly less effective for very "loosened", i.e. low $\tau$ values). For now however, these figures reflect what we saw in practice in our experiments.
>
> > **Could you provide a plot of the time to compute OT problems (standard or Sinkhorn vs. low-rank) as a function of the number of samples ?**
>
> ⧐ Because Sinkhorn and Low-rank approaches solve slightly different problems, it is not easy to compare them in terms of timing. Note, however, that for the **balanced** such comparisons were proposed for the Linear OT problem in [Scetbon et al'20, Scetbon/Cuturi 22] whereas [Scetbon et al'22] compares them in a quadratic setting (GW) problem.
>
> In the unbalanced case, this is even harder, because there is not a 1-to-1 correspondence between their regularizers, convergence criteria etc...  In other words, below 50k points, it's very easy to find a setup where unbalanced Sinkhorn and unbalanced LR Sinkhorn run roughly in the same time, by tweaking thresholds, regularizers etc... **What we can confidently report is that Unbalanced Sinkhorn basically blows up above 50k points, whereas Unbalanced LR Sinkhorn can be easily run "off the shelf", with 500k points and ranks of $r=10$ and aboveas shown in Fig. 1.
>
> > **Also, the use of unbalanced OT involves two extra parameters  which requires cross-validation. Could the authors give plots of the performance of their biology task as a function of…  This would provide interesting insights on the interpretation of these parameters, and their impact on learning tasks.**
>
> ⧐ All of the results provided in the paper do indeed deal with these two parameters using cross-validation. We did not have the time (nor space left in the pdf) to provide accuracy (surface) plots as a function of these two, but we will prepare them for the camera ready. Since our grid for $\tau$ was quite small, we will expand it to provide a more detailed picture as well. This might indeed be useful for practitioners, to see if there are "jumps" (we did not notice them so far, rather smaller variations).

---

> > ### Comment · Reviewer_GuCe · 2023-08-15
> > **Response**
> >
> > Dear Authors,
> >
> > I thank you for your rebuttal, and for your additional experiment comparing the gain of translation-invariant Sinkhorn into your algorithm. While I understand the concerns of other reviewers, I tend to disagree with their focus on the 'incrementality' of your work. To my mind, this is compensated by the thoroughness of the UOT variants you compute, which are as many tools available for future works of the ML community and beyond. You could have tackled each formulation in a different paper to trade quality of work for quantity of publications, which you did not. I could slightly increase my score to support your work, but I'm not sure it would change the global consensus among all reviewers.
> >
> > Side remark: In your answer to fjNH, you say that unbalanced OT solvers are non-convex. It is a convex problem (like balanced OT and unlinke GW), thus all unbalanced OT solvers are also convex. The non-convexity should only arise from your low-rank asumption.

---

> > > ### Author Response · Authors · 2023-08-15
> > > **We are grateful for your reaction to our rebuttal**
> > >
> > > Many thanks for your supportive comments and for taking the time to read our rebuttal.
> > >
> > > > **I thank you for your rebuttal, and for your additional experiment comparing the gain of translation-invariant Sinkhorn into your algorithm.**
> > >
> > > We are happy that you looked at these new results, and that you found them convincing. We heard your concerns on this point, and we will incorporate them into the draft.
> > >
> > > > **While I understand the concerns of other reviewers, I tend to disagree with their focus on the 'incrementality' of your work. To my mind, this is compensated by the thoroughness of the UOT variants you compute, which are as many tools available for future works of the ML community and beyond. You could have tackled each formulation in a different paper to trade quality of work for quantity of publications, which you did not.**
> > >
> > > We believe this is a natural and important debate, and we thank you for your encouraging comments.
> > >
> > > We also hear some of the reviewers' concerns on incrementality. However, at the end of the day, we think these extensions bring a much needed flexibility, and are always easier to derive in hindsight. They also present implementation challenges. For these reasons we believe there is genuine value and novelty in presenting these results to the community.
> > >
> > > > **I could slightly increase my score to support your work, but I'm not sure it would change the global consensus among all reviewers.**
> > >
> > > If, after reading our rebuttal and other reviewers' comments, you believe that our submission is worth appearing at Neurips and could receive a higher grade, then we believe the right course of action would be to update your score. Even if the paper is rejected, this would be an encouragement to us.
> > >
> > > Moreover, we think it might still play a role because:
> > > - anecdotical evidence points to the fact that grades at Neurips are fairly low this year, and the acceptance bar might be accordingly low,
> > > - some of the reviewers (e.g. **fjNH**) have not yet reacted to our rebuttal and/or we are still waiting for their final words (e.g. **1XVN**); they might change their score, and your update could be very important for the AC /SAC/PC to make a final decision.
> > >
> > > > **Side remark: In your answer to fjNH, you say that unbalanced OT solvers are non-convex. It is a convex problem (like balanced OT and unlinke GW), thus all unbalanced OT solvers are also convex. The non-convexity should only arise from your low-rank asumption.**
> > >
> > > Are you referring to this exchange? (copy/pasted for clarity)
> > >
> > > >> **Although the paper heavily sells low rank optimal transport, isn't the formulation in eqs 5/6 non convex? If so, doesn't it make the exact solution intractable? How to ensure convergence towards a good critical point?**
> > > > Yes, the formulation is indeed non-convex, as is also the case for all original unbalanced solvers we build upon. [...]
> > >
> > > If this is the case, then indeed, because Eqs.5/6 in our draft already include the low-rank constraint, making unbalanced LR-OT formulation non-convex. Did we write elsewhere that unbalanced linear OT (e.g. Sinkhorn or others) was non-convex? If we did this is indeed a mistake.
> > >
> > > Thanks again for taking the time to read our rebuttal.

---

### Official Review · Reviewer_1XVN · 2023-06-26

**Soundness:** 3 good
**Presentation:** 2 fair
**Contribution:** 3 good
**Rating:** 4
**Confidence:** 4

**Summary:**

In this paper, the authors combine two variants of optimal transport (OT) known as unbalanced OT and low-rank OT. While the former relaxes the marginal constraints to ease the modelling rigidities and discard possible outliers from input measures, the latter helps reduce the computational cost of $\mathcal{O}(n^3)$, and therefore, makes OT scalable. In addition to standard OT which quantifies the discrepancy between two measures in the same dimensional space, they also apply this combination to the scenarios when two input measures belong to distinct dimensional spaces, which are Gromov-Wasserstein and Fused-Gromov-Wasserstein. Finally, the authors choose the spatial transcriptomic matching problems to justify the practical usage of their proposed methods.

**Strengths:**

- Originality: This work is a novel combination of two versatile and scalable variants of optimal transport (OT), which are unbalanced OT and low-rank OT.

- Quality: The derivations of algorithm proposed in this work are associated with theoretical proof. The empirical performance of the proposed method is demonstrated via the spatial transcriptomic matching problems.

- Significance: the results in this paper are important to some extent as it allows practitioners to find a low-rank solvers for the problem of quantifying the discrepancy between two arbitrary (not necessarily probability) measures.


**Weaknesses:**

- Originality: Although the combination of unbalanced OT and low-rank OT is novel, key tools and techniques (e.g. reparametrization of low-rank couplings, Dykstra algorithm) used in this paper have been already introduced in [1] and [2]. Thus, I think this work is incremental to some extent. To address this concern, I suggest the authors should highlight the main challenges of solving unbalanced low-rank OT compared to its balanced counterpart more clearly.

- Clarity: The presentation of this paper is not good as there are a lot of notations which are either not carefully defined or define inaccurately (see Requested Changes). Additionally, there are some important parts, namely the initialization of Algorithm 1, which should be presented in this paper rather than merely refer to another paper.

**References**

[1] Meyer Scetbon, Marco Cuturi, and Gabriel Peyré. Low-rank sinkhorn factorization. In International Conference on Machine Learning, pages 9344–9354. PMLR, 2021.

[2] Meyer Scetbon, Gabriel Peyré, and Marco Cuturi. Linear-time Gromov-Wasserstein distances using low rank couplings and costs. ICML, 2022.

**Questions:**

1. In line 95, what is the definition of nonnegative rank of $P$? Please add this definition to the revision of this paper.

2. In equation (2), the term $Q\text{diag}(g)R$ seems to be incorrect. Should it be $Q\text{diag}(1/g)R^{\top}$?

3. In equation (7), are there any differences between $\mathrm{KL}(\cdot,\cdot)$ and $\mathrm{KL}(\cdot | \cdot)$? If any, they should be defined explicitly in the paper.

4. Are there any theoretical guarantees that the tuples $(Q_{k+1},R_{k+1},g_{k+1})$ approximate the solution of the optimization problem in equation (6)?

5. In the proposed algorithms for solving ULOT, ULGW and ULFGW, which values of the rank hyperparameter $r$ should we choose? Would it be as low as possible?

**Requested Changes**:

1. References: In lines 38 and 39, when mentioning the usage of Sinkhorn algorithm in solving OT, the authors should cite more relevant papers, namely [1]. Similarly, in line 78, regarding solving unbalanced OT using entropic regularization, the authors should cite the paper [2].

2. In the definition of cost matrix $C$ in Section 2, the index notation $1\leq i,j\leq n,m$ is inacurrate. It should be changed to $1\leq i\leq n$ and $1\leq j\leq m$.

3. Notations: When introducing the unbalanced OT in equation (3), the authors should explain the notations $\mathrm{KL}$, $\tau_1$ and $\tau_2$ rather than assume that readers implicitly understand. Analogously, the notation $A^{\odot2}$ in equation (4) should be defined explicitly.

4. Typo: in line 96, repamatrization --> reparametrization.

**References**

[1] Khang Le, Huy Nguyen, Quang M Nguyen, Tung Pham, Hung Bui, and Nhat Ho. On robust optimal transport: Computational complexity and barycenter computation. Advances in Neural Information Processing Systems, 2021.

[2] K. Pham, K. Le, N. Ho, T. Pham, and H. Bui. On unbalanced optimal transport: An analysis of sinkhorn algorithm. In ICML, 2020.

**Limitations:**

The limitations are not discussed in this paper.

---

> ### Author Rebuttal · Authors · 2023-08-07
>
> Many thanks for providing so much feedback, and giving 3 **good** grades to the *soundness* / *presentation* / *contribution* of our paper. We hope we can convince you to raise your final score with the answers below.
>
> > **Originality: Although the combination of unbalanced OT and low-rank OT is novel, key tools and techniques (e.g. reparametrization of low-rank couplings, Dykstra algorithm) used in this paper have been already introduced in [1] and [2]. Thus, I think this work is incremental to some extent. To address this concern, I suggest the authors should highlight the main challenges of solving unbalanced low-rank OT compared to its balanced counterpart more clearly.**
>
> ⧐ This is an excellent suggestion, and we will emphasise this more clearly around lines 91.
>
> > **Clarity: The presentation of this paper is not good as there are a lot of notations which are either not carefully defined or define inaccurately (see Requested Changes).**
>
> ⧐ We apologize for this lack of clarity. We have incorporated all the changes you have requested.
>
> > **Additionally, there are some important parts, namely the initialization of Algorithm 1, which should be presented in this paper rather than merely refer to another paper.**
>
> ⧐ Finding an efficient initialization for Alg.1 is something of a research topic in itself. For instance, the ott-jax toolbox (https://github.com/ott-jax/ott/blob/main/src/ott/initializers/linear/initializers_lr.py) implements 4 different ways to do this. We will mention this line of work.
>
> > **In line 95, what is the definition of nonnegative rank of ? Please add this definition to the revision of this paper.**
>
> ⧐ Yes, the definition is.that in [Scetbon21,22], we will provide it again.
>
> > **In equation (2), the term [...]**
>
> ⧐ Yes, we apologize for these two bad typos
>
> > **In equation (7), are there any differences between KL? If any, they should be defined explicitly in the paper.**
>
> ⧐ We're sorry about this typo. These 3 terms all refer to the generalised KL computed either for vectors of matrices with positive entries, $ \textrm{KL}((p,q) = \sum_i p_i \log \frac{p_i}{q_i} -p_i +q_i$. We will switch to $\textrm{KL}(\cdot|\cdot)$ for all.
>
> > **Are there any theoretical guarantees that the tuples approximate the solution of the optimization problem in equation (6)?**
>
> ⧐ Such guarantees cannot be achieved, to our knowledge, because the problem is at least as non-convex as the balanced case.
>
> > **In the proposed algorithms for solving ULOT, ULGW and ULFGW, which values of the rank hyperparameter should we choose? Would it be as low as possible?**
>
> ⧐ In all experiments, we have used cross-validation on train sets to set rank $r$. Lower $r$ means faster computations, but they might end up in factorized transports that are too coarse to perform well. We believe $r$ should be set depending on the problem size vs. compute ability, much like k-means (or, to some extent, entropic regularization).
>
> > **References: In lines 38 and 39, when mentioning the usage of Sinkhorn algorithm in solving OT, the authors should cite more relevant papers, namely [1]. Similarly, in line 78, regarding solving unbalanced OT using entropic regularization, the authors should cite the paper [2].**
>
> ⧐ Thanks for these two very relevant references, and we apologize for having missed them, we have added them to our draft.
>
> > **In the definition of cost matrix in Section 2, the index notation [...]**
>
> ⧐ Sure, we happy to change this.
>
> > **Notations: When introducing the unbalanced OT in equation (3), the authors should explain the notations KL in equation (4) should be defined explicitly.**
>
> ⧐ We updated our paper, and followed your advice to clarify these notations from the start (KL and Hadamard exponent)
>
> > **Typo: in line 96, repamatrization --> reparametrization.**
>
> ⧐ We had a typo indeed, thanks! It seems reparamet**e**rization is more common, so we will use it.

---

> > ### Comment · Reviewer_1XVN · 2023-08-11
> >
> > Dear authors,
> >
> > Thanks for making great effort to do the rebuttal, I really appreciate that. However, my following two concerns havenot been addressed yet. In particular,
> >
> > 1) Although the combination of unbalanced OT and low-rank OT is novel, key tools and techniques (e.g. reparametrization of low-rank couplings, Dykstra algorithm) used in this paper have been already introduced in [1] and [2]. Thus, I think this work is incremental to some extent.
> >
> > 2) What are the main challenges of solving unbalanced low-rank OT compared to its balanced counterpart?
> >
> > Additionally, I would like to apologize for making a mistake of grading the presentation score. As I said in the Weaknesses section, the presentation is not good as there are a lot of notations which are either not carefully defined or define inaccurately. Thus, I reduce the presentation score to 2, but still keep the current rating of 4 given your response in the rebuttal.

---

> > > ### Author Response · Authors · 2023-08-12
> > > **Thanks for your prompt reaction**
> > >
> > > Thanks for your reaction. Let us clarfiy your final concerns.
> > >
> > > 1. We hear you when you say that you feel our work is `incremental to some extent`. We do note, however, that you have also graded the "contribution" part of our paper as **"good" (3)**, which indicates that, in a bigger picture, you are satisfied about the contributions themselves to a larger extent.
> > >
> > >    Reviewer's `GuCe` statement summarizes very well our position: `The contributions might look incremental at first (combining unbalanced OT with low-rank OT), but are actually very thorough. They do not restrict to this one combination, but consider all recent variants of unbalanced OT (Unbalanced OT, GW and Fused GW) which have been developped in the literature in the last years.`
> > >
> > >    As a community, a question worth asking is whether (1) the OT toolkit would benefit from an unbalanced formulation for low-rank solvers, which are taking an increasingly important role (2) the community would benefit from a solid algorithmic reference with exhaustive experiments in large scale settings that presents this.
> > >
> > >    The reason we carried out all of these generalizations was motivated by several requests from practitioners to push into this direction, as summarized in Lines 51-54. **Therefore we believe your impression that the paper is "incremental to some extent" should be balanced with the timeliness, technicality and exhaustiveness of our proposal, as well as the wealth of experiments we have proposed.**
> > >
> > > 2. We understand your concern. There are always computational challenges when computing regularized OT, and quite often "hidden" problems. However, let us clarify in very clear terms that **there were no unexpected challenges, computationally and practically speaking, that have arisen *because* of our unbalanced generalizations**, compared to using LR OT or LR GW in general. While these tools can be challenging to use, our paper does not introduce "new challenges".
> > >
> > >     The only obvious practical challenge is of course that of choosing regularizer strength, but this is the case for *all* unbalanced formulations, low-rank or not. We handled it as rigorously as we could, using cross-validation in all our experiments, in a way that is, in our opinion, more transparent than in most previous papers. Results have been positive on that front.
> > >
> > >     The only computational issue that *might* have appeared would have arisen from the translation invariance problem, but we took care of it. This leads to clear improvements, as shown in our rebuttal pdf.
> > >
> > >     **We hope this alleviates your second concern.**  As usual, the ultimate answer on this will be to hear from practitioners that will be able to test our tool, once we open-source it. In the meantime, we believe Neurips would be the perfect venue to advertise properly these generalizations.
> > >
> > >     We are happy to add anything to our paper that you feel we might have missed on that front, but we believe our current writing reflects accurately the message above.
> > >
> > >
> > > As for presentation, we do acknowledge a few minor typos that have peppered the paper (notably inconsistency in $KL(\cdot | \cdot)$ vs. $KL(\cdot, \cdot)$ or $g\rightarrow 1/g$ and $R\rightarrow R^T$ in the first equation), but this has been fixed in a few minutes. We believe this is where the value of reviewing comes from, and we are grateful for your help/time in this regard.
> > >
> > > We sincerely appreciate the time you have put into reading our rebuttal.

---

> > > > ### Comment · Reviewer_1XVN · 2023-08-12
> > > >
> > > > Thanks for your response. In this thread, I will explain for my grading, and discuss the concerns I raised previously. However, I would like to remind you of respect for reviewers as I do not feel it in your previous response.
> > > >
> > > > Firstly, I graded your contribution as good (3) since you introduced a new combination of unbalanced and low-rank optimal transport associated with theoretical and empirical guarantee. And the only reason you did not get a 4 for it was that the proof techniques were quite straightforward from previous work [1, 2]. Therefore, I asked you about the (theoretical and empirical) main challenges of unbalanced settings compared to balanced ones. But you had not answered this question until I asked it again in the discussion phase. Then, you confirmed the triviality of adapting techniques from balanced settings to unbalanced settings in your second response. Thus, I doubt whether this paper really deliver a good contribution or not.
> > > >
> > > > Secondly, I should have graded your presentation as fair (2) at the first place, but somehow I made a mistake by grading it 3 as I clarified in my previous response. This was not only because of numerous typos but also due to many undefined terms, namely the nonnegative rank of a matrix, $A^{\odot 2}$ in equation (4), $\tau_1$ and $\tau_2$, etc. Additionally, I also agree with other reviewers that the experiment section is unreasonably brief and not sufficiently discussed while there are some unnecessary presentations of algorithms in the main paper. Given those reasons, I think the presentation score of 2 is reasonable.
> > > >
> > > > Based on what we have discussed so far, I decide to remain the rating of this paper.
> > > >
> > > > Thank you,
> > > >
> > > > **References**
> > > >
> > > > [1] Meyer Scetbon, Marco Cuturi, and Gabriel Peyré. Low-rank sinkhorn factorization. In International Conference on Machine Learning, pages 9344–9354. PMLR, 2021.
> > > >
> > > > [2] Meyer Scetbon, Gabriel Peyré, and Marco Cuturi. Linear-time Gromov-Wasserstein distances using low rank couplings and costs. ICML, 2022.

---

> > > > > ### Author Response · Authors · 2023-08-12
> > > > > **communication issues**
> > > > >
> > > > > Dear Reviewer
> > > > >
> > > > > It seems there are a few communication issues. We have kept a respectful tone during this discussion.
> > > > >
> > > > > Ultimately, we feel your score does not reflect the weaknesses you have pointed out in our paper, that are mostly a question of form, and not questioning messaging, content or experimental validation. We mean no disrespect but need to point out this inconsistency.
> > > > >
> > > > > > **But you had not answered this question until I asked it again in the discussion phase.**
> > > > >
> > > > > We hope the rebuttal helps, by definition, reach a consensus iteratively. We had trouble understanding exactly what you meant in your former question.
> > > > >
> > > > > > **Then, you confirmed the triviality of adapting techniques from balanced settings to unbalanced settings in your second response.**
> > > > >
> > > > > This is a major misunderstanding of what we wrote, and we would be grateful if you could avoid using such a contentious word as “triviality” in the context of this rebuttal.
> > > > >
> > > > > Your question, in the latest response, was on the challenges of **solving** unbalanced OT
> > > > >
> > > > > > **What are the main challenges of solving unbalanced low-rank OT compared to its balanced counterpart?**
> > > > >
> > > > > Our answer was that, thanks to our algorithms, solving the unbalanced formulation is now reasonably easy, and does not introduce more challenges than what a practitioner would expect from solving a large scale LR OT problem (i.e. some hyper-parameter tuning, basic experimentation with initializers). It seems you misunderstood our answer as stating that we claimed our work was trivial.
> > > > >
> > > > > We now understand your question as "was what kind of challenges **we, as authors, encountered on the way to proposing an unbalanced formulation of LR-OT**" Of course, we can detail this, this looks a lot like summing up the difficulties we encountered coming up with our results. We spend most of our time working towards:
> > > > > - being able to generalize to an LR setting all the unbalanced ideas introduced in former proposals for non-low rank problems, while still being able to retain the favorable linear computational complexities of LR approaches, even in Fused cases;
> > > > > - formulate algorithms that could be close enough to former work, to make them provably convergent and benefit from known initialization techniques for LR;
> > > > > - implemented these complex algorithms properly (some of our formulas end up being technically quite heavy), write them up;
> > > > > - gather all of these large scale datasets and find a proper scaling of hyperparameters that would run "off the shelf" on new problems (as we demonstrated by running the quantized GW task requested by reviewer **ckny** in a few hours).
> > > > >
> > > > > We hope we have lifted an important misunderstanding in our discussion, and we are sorry that the rebuttal format is prone to such mistakes.

---

### Official Review · Reviewer_ckny · 2023-06-28

**Soundness:** 3 good
**Presentation:** 3 good
**Contribution:** 2 fair
**Rating:** 4
**Confidence:** 3

**Summary:**

The authors proposed unbalanced low-rank OT (ULOT) solver, which is an extension of the balanced counterpart. They show how to adapt this solver to the other unbalanced low-rank settings, namely translation-invariant and GW. The experiments compare the performances of multiple low-rank solvers, and of the Unbalanced Fused GW.

**Strengths:**

The extensions from the balanced to unbalanced LOT, as well as from ULOT to ULFGW are not trivial and require some calculation effort. I also find that the writing is very instructive and all technical details are clearly presented and on point.

**Weaknesses:**

It seems to me that
- The content in the main paper and appendix is not adequately partitioned, namely the section 3 is somewhat too long that there is few space left for the experiment section, and some experiment details in the Appendix should be put in the main.
- The main contribution of the paper is quite incremental.

More precisely,

+ While adding translation-invariant improves the convergence of the usual unbalanced OT, it is unclear about the real improvement in the experiments. IMHO, it is more or less an add-on of the ULOT solver and should be moved to Appendix because it would dilute the main message of the paper (which is about ULOT). It is enough that Algo 4 and 5 use the ULR-Dykstra solver (Algo 6), instead of Algo 3.

+ The Algo 4 is merely a special case of Algo 5, where alpha = 0, thus should be removed. While it is more instructive to start with the GW setting before moving to the fused GW one, all the details can be moved to the Appendix and section 3.3 and 3.4 should be restructured or/and merged.

+ It seems that the experiment section is not sufficiently diverse because the experiments and competing methods are restricted to just a family of the low-rank based methods (except FUGW). IMHO, since the contribution on the methodogoly (i.e. section 3) is not very significant, I would love to see more comparison with other methods, like (unbalanced) mini-batch OT, or quantized GW, and maybe on more experiments (e.g. on graphs).

+ I find it weird that the section 4.3 is unreasonably brief and not sufficiently discussed, while most of its details are moved to the Appendix. This makes the section 4.3 look incomplete in the main paper.

**Questions:**

In Algorithm 2: typo in 1 / gamma / tau1


**Limitations:**

The authors do not discuss the limitations of their work.

---

> ### Author Rebuttal · Authors · 2023-08-07
>
> Many thanks for the wealth of comments you have provided.
>
> > **The content in the main paper and appendix is not adequately partitioned, namely the section 3 is somewhat too long that there is few space left for the experiment section, and some experiment details in the Appendix should be put in the main.**
>
> ⧐ Thanks. We agree with your suggestion and have removed many technical details from the main section to the appendix to free up space for experiments.
>
> > **While adding translation-invariant improves the convergence of the usual unbalanced OT, it is unclear about the real improvement in the experiments.**
>
> ⧐ We agree that this was missing. We have now added such comparisons, showing a clear-cut benefit when using our TI variant. Please look at our Pdf, Figure X **[TODO Michal]**
>
> > **The Algo 4 is merely a special case of Algo 5, where alpha = 0, thus should be removed. While it is more instructive to start with the GW setting before moving to the fused GW one, all the details can be moved to the Appendix and section 3.3 and 3.4 should be restructured or/and merged.**
>
> ⧐ As you mention, we felt it was instructive to start with GW. While Alg. 4 is a particular case of Alg. 5, we believe the most important conceptual gap lies in going from 3 to 4. We prefer following your recommendation to free up space by moving Alg. 5 (ULFGW) to the appendix.
>
> > **It seems that the experiment section is not sufficiently diverse because the experiments and competing methods are restricted to just a family of the low-rank based methods (except FUGW). IMHO, since the contribution on the methodogoly (i.e. section 3) is not very significant, I would love to see more comparison with other methods, like (unbalanced) mini-batch OT, or quantized GW, and maybe on more experiments (e.g. on graphs).**
>
> ⧐ Thanks for your suggestions, here are our answers:
>
> - **It is not clear to us why and how mini-batch OT could be a competitor to handle the large scale unbalanced GW problems we solve**. Mini-batch OT was introduced to *minimize* a _linear_ OT _loss_ (e.g. [FZFGC] Fatras, K., Zine, Y., Flamary, R., Gribonval, R., & Courty, N. (2019). *Learning with minibatch Wasserstein: asymptotic and gradient properties*), **to train typically a generative model**. There, the goal is to minimise a fitting term $\mathcal{L}(\theta)=W(p_\theta, q),$ and both (or either) $p_\theta$ and $q$ are continuous densities. The argument in favor of mini-batch is that sampling many times from $p_\theta,q$ and/or $q$ to average multiple gradients for $\theta$ is better than using the gradient computed from a single bigger batch.
>
>    All our experiments consider instead two **fixed**, large datasets of point-clouds, in a _quadratic_ GW setting. Here a mini-batch approach would result in an arbitrary partitioning of each measure into sub-measures, and then match arbitrarily these smaller sub-measures. This would be very naive, suboptimal, and we are not aware of papers advocating this except when doing more elaborate hierarchical approaches (e.g. MREC, https://arxiv.org/abs/2001.01666). MREC was compared to LRGW (see [Scetbon+22]), but does not have an unbalanced formulation.
>
> - **Quantized GW is not an unbalanced method**, therefore we feel it would be out of place in this study. Yet, we have considered one of their experiments (the scene dataset) in our new experiments (see pdf in global response). The authors of Quantized GW provide very little details on their experimental setup (hyperparameter tuning in particular) hence we did not re-run their solver. We only used their reported accuracy and timing in the text (which we assume is the best they could get). With the limited time of this rebuttal, we get roughly the same accuracy as shown in our pdf using ~3 min of compute.
>
> - Note that all experiments in 4.3 compare two brain meshes, i.e. two graphs endowed with shortest path metrics.
>
> To summarize our thoughts, we believe that the natural competitor for our setting remains entropic unbalanced solvers, because:
> - they are by far the most studied alternative, having been used in dozens of papers;
> - they can be used in an unbalanced setting in a fairly comparable way to ours;
>
> > **I find it weird that the section 4.3 is unreasonably brief and not sufficiently discussed, while most of its details are moved to the Appendix. This makes the section 4.3 look incomplete in the main paper.**
>
> You are right. Because we were short of space at the time of submission, we mostly reported results and not experimental details (the benchmark is not ours). However, this can be easily corrected as we can expand on the material in our supplementary and bring it back to the main body. Thanks to the space freed by Algorithm 5, we can now rebalance this section, and move back relevant background in 4.3.

---

> > ### Comment · Reviewer_ckny · 2023-08-13
> > **Response to the authors**
> >
> > I thank the authors for their response. It helps clarify most of my concern. I do agree that the two suggested competing methods (minibatch OT and quantized GW) don't really fit in the context of the paper.
> >
> > However, comparing to the prior work on LOT and low-rank GW, I would still consider the contribution of ULOT (and its extensions) to be incremental. I see it as a mere stack of many prior works (low rank, unbalanced, GW) and there has not yet been any theoritical analysis of the proposed method in the paper. Even if it is possible that some results in the prior works can be applied, I'm not convinced that it is enough for a real contribution, except something really stands out. For this reason, I expect to see more contribution on the application side to compensate for the lack of theoritical understanding.
> >
> > In conclusion, I decide to increase the score to 4, while still being more prone to the reject.

---

> > > ### Author Response · Authors · 2023-08-14
> > > **Thanks for reading our rebuttal**
> > >
> > > Dear Reviewer,
> > >
> > > Many thanks for reacting to our rebuttal. We are grateful that you have raised your score despite your remaining concerns.
> > >
> > > We understand your impression that our work may seem incremental, but we hope that you can agree that:
> > > - there are many technical difficulties in our work. Even if you were to pick the most informed people in the OT community, we do not think that they would be able to carry out the computations we did without effort.
> > > - our goal was to answer various requests from users that see in LR approaches the key to scale up OT to very large datasets, but cannot yet, as we explained in Lines 51, use an unbalanced setting. This is in itself real problem for them.
> > >
> > > For these reasons, we believe there is merit in publishing these works.
> > >
> > > We also understand that you felt the experimental section was a bit rushed. However, we believe that there is already a lot of content in our experiments, notably if we add the few remaining bits provided in the attached pdf. Notice that:
> > > - Our spatial-transccriptomics datasets are among the largest we have seen tested for a FGW setting.
> > > - **You are perfectly right** when you claim that Section 4.3 is too short, and does not contain more details. Your concern was heard, and we are expanding that part. The take home message, however, is that our method slightly outperforms the method proposed in [https://arxiv.org/pdf/2206.09398.pdf], presented at Neurips last year, on the same dataset.
> > >
> > > Thanks again for reading our rebuttal, and for your detailed review.
> > > the authors

---

### Official Review · Reviewer_fjNH · 2023-07-10

**Soundness:** 3 good
**Presentation:** 3 good
**Contribution:** 2 fair
**Rating:** 6
**Confidence:** 4

**Summary:**

In its original exact formulation, optimal transport (OT) suffers from an $\mathcal{O}(n^3)$ cost when applied to clouds of $n$ points. In addition, the exact constraint on the marginals makes it sensitive to outliers.
One solution for each of these problems exist:
- unbalanced OT [Schiebinger et al., 2019] is less sensitive to outliers than regular OT, as it relaxes the marginal constraints into a KL-penalized form
- entropic OT allows for the use of the celebrated Sinkhorn algorithm but still requires large matrix vector multiplications. Low rank OT  [Forrow et al., 2018] is a promising direction to further improve upon entropic OT, by using a low rank cost matrix and low rank constraints of the plan.

Building on a these two seminal papers and a line of work by Scetbon, Cuturi and coauthors, the paper combines low rank OT with unbalanced OT (Formulation in Eq 5).
A dedicated algorithm is proposed, interpreted as proximal gradient descent in the KL geometry. The latter is improved based on the works of Séjourné et al [2022].

The approach is extended to the Gromov-Wasserstein and Fused Gromov-Wasserstein OT problems.

**Strengths:**

OT has found many applications in practice in the last decade; alleviating its cost extends its range of applications. The proposed formulation is sound, the treatment is relatively easy to follow and the derivations are made clear.

**Weaknesses:**

- Although the paper heavily sells low rank optimal transport, isn't the formulation in eqs 5/6 non convex? If so, doesn't it make the exact solution intractable? How to ensure convergence towards a good critical point?
- Convergence of the algorithm
    - Known results about Dykstra's algorithm are used to compute the solution of Equation 7 via its dual. Can the authors provide a reference for the convergence of the whole algorithm, namely the iterations defined by 7? I may have missed it, but I could not find reference of convergence towards a critical point;
    - convergence seems to be measured in terms of difference between two successive iterates going to zero. It is well-known that this does not guarantee convergence of the iterates (e.g. take the harmonic series). To clarify, do the author have a classical convergence result?

**Questions:**

- In the last cost of eq(2) I believe it should be diag(1/g) not diag(g), and $R^T$ not $R$


Typos:
- reparamaterized, repamatrization
- mass conversation for mass conservation
- misuse of \cite or \citep vs \citet , e.g. "of [Frogner et al., 2015, Chizat et al., 2018]", and several other places e.g. "notations from [Scetbon et al., 2021]"
-  towards a stationary points
- such that with probability at least 0.99 that: extra "that"
- Algo 1: (check me)
- paper uses \star and * for $\lambda^\star$

---

> ### Author Rebuttal · Authors · 2023-08-07
>
> Many thanks for your time and for your comments.
>
> > **Although the paper heavily sells low rank optimal transport, isn't the formulation in eqs 5/6 non convex? If so, doesn't it make the exact solution intractable? How to ensure convergence towards a good critical point?**
>
> Yes, the formulation is indeed non-convex, as is also the case for all original unbalanced solvers we build upon.
>
> While this removes several theoretical guarantees, practice suggests a “mild” form of non-convexity, comparable to e.g. k-means or non-negative matrix factorisation (NMF).
>
> Convergence to a critical point is ensured through Dykstra, but the solution depends on the initialization. We use the standard approaches considered for the balanced case (see e.g. https://github.com/ott-jax/ott/blob/main/src/ott/initializers/linear/initializers_lr.py )
>
> > **Can the authors provide a reference for the convergence of the whole algorithm, namely the iterations defined by 7? I may have missed it, but I could not find reference of convergence towards a critical point;**
>
> Sure, in [Scetbon et al. , 2021]  "Low-rank sinkhorn factorization", the authors obtain in Appendix A a generic approach to show the stationary convergence of the mirror-descent scheme as soon as the objective function to minimize is relatively smooth w.r.t the negative entropy. Also the authors show the constrained version of the objective defined in (5) is relatively smooth w.r.t the negative entropy. Therefore the stationary convergence of our algorithm is a corollary of the result obtained in [Scetbon et al. , 2021]. We will clarify this point in the paragraph on "Convergence and Complexity" written page 4.
>
> > **convergence seems to be measured in terms of difference between two successive iterates going to zero. It is well-known that this does not guarantee convergence of the iterates (e.g. take the harmonic series). To clarify, do the author have a classical convergence result?**
>
> For convex OT problems (e.g. Sinkhorn), the usual approach is to compute the gradient norm (marginal violations in balanced case) to monitor convergence. For nonconvex OT problems (unbalanced solvers but also Gromov-Wasserstein and Wasserstein barycenter problems with free support), it is more usual to monitor changes in the solution (as in, e.g., k-means).
>
> Note also that our iterates are all bounded (they are transport plans $Q,R$ and simplex vector $g$), there is therefore no "divergence" we might miss (as in your harmonic series example), and we have never seen such behaviour in practice.
>
> > **In the last cost of eq(2) I believe it should be diag(1/g) not diag(g) [...]**
>
> Indeed, we apologize for this unfortunate typo early in the paper. Thanks for pointing it out.
>
> > **reparamaterized, repamatrization**
>
> Fixed! we will use reparam**e**terized and reparameterization (https://en.wiktionary.org/wiki/reparameterize#English ).
>
> > **mass conversation for mass conservation**
>
> We apologize for this typo, likely due to an unfortunate auto-correct.
>
> > **misuse of \cite or \citep vs \citet , e.g. "of [Frogner et al., 2015, Chizat et al., 2018]", and several other places e.g. "notations from [Scetbon et al., 2021]”**
>
> To clarify: we use `\citep` to refer to the *paper*, and `\citet` to refer to the authors of the work. We meant therefore: “the framework of [these papers]” and “notations from [this paper]”. When crediting a group of authors we use `\citet`, as in line 132 (`Séjourné et al. [2022] have proposed`). We have found a few mistakes disagreeing with this convention, we have fixed them.
>
> > **towards a stationary points, such that with probability at least 0.99 that: extra "that", Algo 1: (check me), paper uses \star and * [...]**
>
> Many thanks for pointing out these typos and unfortunate "left over" comments.

---

> > ### Comment · Reviewer_fjNH · 2023-08-21
> >
> > The authors' rebuttal and comments from reviewer `GuCe` on the non triviality of the combination of techniques, together with the extent of the addressed problem, make me increase my score. I thank the authors for their precise answers.

---

> > > ### Author Response · Authors · 2023-08-21
> > > **Many thanks for reading our rebuttal**
> > >
> > > We are grateful for your comments, for catching these typos, and for your questions. We will use them to improve our draft. We are very thankful for your score increase.

---

### Author Rebuttal · Authors · 2023-08-09

**We thank the reviewers, AC and SAC assigned to this paper for their time and work looking into our submission**.

We thank them in advance for reading our rebuttal and interacting with us for a few more days during the discussion period.

We were happy to see that the paper was overall well received by all 4 reviewers:

**fjNH** : *The proposed formulation is sound, the treatment is relatively easy to follow and the derivations are made clear.*

**ckny**: *The extensions [...] are not trivial and require some calculation effort [...] the writing is very instructive and all technical details are clearly presented and on point.*

**1XVN**: *the results in this paper are important to some extent[...]*, *This work is a novel combination*

**GuCe**: *The contributions might look incremental at first (combining unbalanced OT with low-rank OT), but are actually very thorough.*

The most important weaknesses highlighted by reviewers point to:
- some sloppiness in our presentation: we proposed several tightly related, yet distinct algorithms. This led to some redundancy, as discussed below in our answers. Because of a lack of space, some of the experimental sections were left to their minimum, with only (good!) results presented. As mentioned by reviewer **ckny**, this is true in particular for Section 4.3.

➡ *We have incorporated comments by reviewers in our draft, and have removed an algorithm from the main body to the appendix, and added back experimental details.*

- some clarifications in experiments, e.g. importance of using the translation-invariant adjustment, and comparing to quantized GW (which is not, however, an unbalanced method).

➡ *We have run novel experiments following their remarks. They all reinforce the message of our submission, and will be added either to appendix or main body, depending on space.*

- a few "bad" typos (e.g. $D(g)$ instead of $D(1/g)$ and $R$ instead of $R^T$).

➡ *We apologize for these unfortunate typos, we have corrected them.*

We believe we have addressed all of the points raised by reviewers. In addition, we will release code for *all* of the tools presented in the paper in one of the major OT python toolboxes. Our implementation can run on CPU but also natively on GPU. We will share the code before the end of August.

At the moment, all reviewers have scored our paper as **good (3)** in *both* **soundness** and **presentation**. We have received an average 2.75 grade (2,2,3,4) in **contribution**, and we hope our rebuttal alleviates these concerns.

We believe that the very supportive words found in all reviews are not reflected in the current distribution of (fairly low) scores of 7,4,4,3 . If reviewers agree with our assessment, we humbly ask them to reconsider their score.

---

### Comment · Area_Chair_c1HP · 2023-08-11

 Dear reviewers and authors,

Thank you very much for your work on this submission and its evaluation. Now that the authors have responded to the reviews, I **strongly encourage** the reviewers to acknowledge the review, to look at other reviews and rebuttals for this submission, and to adjust their scores if needed. Thanks to those that have already done so.

Authors have the possibility to reply if further questions are needed, until the 16th.

Thank you very much to all,

Area Chair

---

### Decision · Program_Chairs · 2023-09-21

**Decision:**

Accept (poster)

**Comment:**

The paper proposes a solution to the problem of efficiently compute discrete unbalanced Optimal Transport (OT) problems.
By assuming that the transport plan is a low-rank matrix, they leverage on this property to reduce
complexity of the global algorithm. The authors extend this to unbalanced GW and unbalanced Fused Growov

It is a sound and well built paper.

From my reading of the paper and discussions with the reviewer and SAC, it seems that the ideas proposed in the paper are
indeed incremental in several of their aspects albeit well executed,

* Section 3.1 borrows heavily on Scetbon et al. (2021, 2022)
* Section 3.2 is an extension of Sejourné 2022 Proposition 2
* Section 3.3 is an extension of Scetbon et al (2022)
* Section 3.4 is a mix of 3.1 and 3.3 and the algorithm subsumes the one in Section 3.3

However the methodology can indeed be useful for practictioners.

Given this very borderline situation, after many discussions, we have decided on acceptance, given the
full batch of papers of the SAC and the resulting calibration.